# PD-L1 signaling selectively regulates T cell lymphatic transendothelial migration

Wenji Piao [1,2✉], Lushen Li[1,2], Vikas Saxena [1,2], Jegan Iyyathurai[1], Ram Lakhan[1,2], Yigang Zhang[3], Isadora Tadeval Lape[4], Christina Paluskievicz[2], Keli L. Hippen[3], Young Lee[2], Emma Silverman[1], Marina W. Shirkey[1], Leonardo V. Riella[4], Bruce R. Blazar[3] & Jonathan S. Bromberg[1,2,5✉]

Programmed death-1 (PD-1) and its ligand PD-L1 are checkpoint molecules which regulate immune responses. Little is known about their functions in T cell migration and there are contradictory data about their roles in regulatory T cell (Treg) function. Here we show activated Tregs and CD4 effector T cells (Teffs) use PD-1/PD-L1 and CD80/PD-L1, respectively, to regulate transendothelial migration across lymphatic endothelial cells (LECs). Antibody blockade of Treg PD-1, Teff CD80 (the alternative ligand for PD-L1), or LEC PD-L1 impairs Treg or Teff migration in vitro and in vivo. PD-1/PD-L1 signals through PI3K/Akt and ERK to regulate zipper junctional VE-cadherin, and through NFκB-p65 to up-regulate VCAM-1 expression on LECs. CD80/PD-L1 signaling up-regulates VCAM-1 through ERK and NFκB-p65. PD-1 and CD80 blockade reduces tumor egress of PD-1[high] fragile Tregs and Teffs into draining lymph nodes, respectively, and promotes tumor regression. These data provide roles for PD-L1 in cell migration and immune regulation.

[1] Center for Vascular and Inflammatory Diseases, University of Maryland School of Medicine, Baltimore, MD 21201, USA. [2] Department of Surgery, University of Maryland School of Medicine, Baltimore, MD 21201, USA. [3] Division of Blood & Marrow Transplant & Cellular Therapy, Department of Pediatrics, University of Minnesota Cancer Center, Minneapolis, MN 55455, USA. [4] Center for Transplantation Sciences, Department of Surgery, Massachusetts General Hospital, Boston, Massachusetts, Boston, MA 02114, USA. [5] Department of Microbiology and Immunology, University of Maryland School of Medicine, Baltimore, MD 21201, USA. ✉email: wpiao@som.umaryland.edu; jbromberg@som.umaryland.edu

PD-1 and its ligands PDL-1/2 attenuate immune responses[1,2]. PD-1 is induced on activated lymphocytes, dendritic cells (DCs), and monocytes[3,4]. PD-L1 is widely expressed in diverse cells including vascular endothelial and tumor cells[1,5–7], while PD-L2 is restricted to DCs and macrophages. During infection, DC PD-L1 interacts with PD-1 on Teffs to induce T cell anergy and exhaustion[8–11]. PD-1$^{-/-}$ and PD-L1$^{-/-}$ mice develop spontaneously and exacerbated autoimmunity, respectively[2,10,12].

Tregs, characterized by the expression of transcription factor forkhead box protein P3 (FoxP3), can be divided into natural thymus-derived Tregs (tTregs) and peripherally induced Tregs (iTregs). PD-1 expression correlates inversely with Treg expansion and immune suppression in vivo[13]. In livers of patients with chronic virus infection, Tregs have higher PD-1 expression than circulating Tregs or intrahepatic Teffs[13]. Likewise, activated Tregs in tumor-infiltrating lymphocytes (TILs) express higher levels of PD-1 than circulating Tregs. The ratio of Tregs to Teffs in TILs is significantly decreased after anti-PD-1 monoclonal antibody (mAb)-treatment[14], and is attributed to enhanced Teffs and decreased Tregs[15]. PD-L1 ligation to PD-1 leads to different functional outcomes: binding to effector T cells suppresses TCR or costimulatory signaling and causes apoptosis, anergy, and exhaustion[16]; while binding to Tregs leads to sustained Foxp3 expression and proliferation[17,18]. On the other hand, PD-1 deficiency or blockade can enhance the immunosuppressive activity of TIL Tregs and promote tumor growth[14,19], and PD-1$^{-/-}$ Tregs are highly proliferative and immunosuppressive[14,19–21]. These seemingly contradictory observations about the role of PD-1 in Treg function suggest PD-1 may regulate other aspects of Tregs.

PD-L1 is constitutively expressed on LECs, and is increased in inflamed tissues or the tumor microenvironment (TME)[22,23]. In lymph nodes (LNs), PD-L1 is most highly expressed by LECs compared to other stromal cells such as blood endothelial cells or fibroblastic reticular cells[24]. Since LECs are important for TEM of lymphocytes, these observations raise the possibility that LEC PD-L1 not only directly regulates lymphocyte activation at sites of inflammation but may also regulate other functions such as migration.

In the present study we found activated Tregs expressed the highest level of PD-1 among different T cell subsets. This prompted us to investigate if Treg PD-1 interacted with LEC PD-L1 to regulate other aspects of Treg function. Our investigations uncovered another role for the PD-1-PD-L1 interaction in Treg migration across LECs and into the draining lymphatics.

## Results

### Differential expression of PD-1 and PD-L1 by T cells and LECs.
Among resting CD4 T cell subsets, murine Foxp3GFP$^+$CD4$^+$CD25$^+$CD44$^{low}$ tTregs expressed both PD-1 and PD-L1 and the levels were higher compared to Foxp3GFP$^-$CD25$^-$CD4$^+$CD44$^{low}$ naïve CD4 T cells. Among activated CD4 T cell subsets, Foxp3GFP$^+$CD25$^+$CD4$^+$CD44$^{high}$ iTregs and Foxp3GFP$^-$CD25$^+$CD4$^+$CD44$^{high}$ activated CD4 Teffs (effector phenotype and function shown in Supplementary Fig. 1a, b) expressed even higher levels of both molecules, with iTregs expressing the highest level of PD-1 of all subsets but lower levels of PD-L1 than Teffs (Fig. 1a). Human CD4$^+$CD25$^{high}$CD127$^-$CD45RA$^-$ activated naïve Tregs which resemble iTregs[25,26] also had higher PD-1 and lower PD-L1 expression than human CD4$^+$CD25$^-$CD127$^+$CD45RA$^+$ CD4 effector T cells (Fig. 1b). Notably, CD80, another ligand of PD-L1, was expressed at higher levels on the mouse and human Teffs than activated Tregs (Fig. 1a, b). Naïve CD8 T cells and B cells expressed negligible levels of

PD-1, PD-L1, or CD80, while activated CD8 T cells had a marked increase in PD-1 expression, and only a modest increase in PD-L1 and CD80. Activated B cells had major (40–100-fold) increases in PD-1 and CD80, but no change in PD-L1 expression (Supplementary Fig. 2a, b). Immature bone-marrow-derived DC (imBMDC) expressed PD-1, PD-L1, and CD80. All three molecules were further enhanced after imBMDC maturation (Supplementary Fig. 2c).

PD-L1 was highly expressed on both mouse and human LECs, while minimal PD-1 was detected on the surface of the LECs (Fig. 1c). Immunohistochemistry and flow cytometry analysis of permeabilized LECs showed intracellular PD-1 expression while PD-L1 was predominantly on the cell surface (Fig. 1d). To determine PD-L1 expression on LEC in physiological conditions, fresh primary dermal LECs (Lyve-1$^+$CD31$^+$) were isolated and found to have comparable levels of PD-L1 expression as the cultured primary LECs (Fig. 1c). Minimal levels of PD-L1 or PD-1 were detected on Lyve-1$^-$CD31$^-$ non-endothelial cells.

### Tregs and Teffs differentially engage PD-L1 on LECs for TEM.
We previously validated our model of lymphatic TEM, using primary mouse LEC monolayers grown on the membrane of a Boyden chamber and measuring the migration of various leukocyte subsets from the basal (abluminal) to the apical (luminal) side of the endothelial cells in response to a chemokine gradient[25,27]. Using this assay, blocking PD-1 with anti-PD-1 mAb (Rmp1-14) on mouse iTregs or tTregs inhibited TEM across LECs toward CCL19, a potent T cell chemokine, in a dose-dependent fashion (Supplementary Fig. 3a, b). In contrast, blocking PD-L1 on Tregs with anti-PD-L1 mAb (10 F.9G2) did not inhibit TEM (Fig. 2a). Blocking PD-1 or PD-L1 on Teffs or naïve CD4 did not inhibit TEM (Fig. 2b, Supplementary Fig. 3c, d). No inhibition of Treg TEM was observed when they migrated across the plastic membrane of the Boyden chamber without LECs (Fig. 2a, b), suggesting the interaction of Treg PD-1 with PD-L1 on LECs was essential for regulating migration. To test the possible effect of antibody-FcR interactions or antibody cross-linking on the blockade, we generated (Fab')2 and Fab fragments. Both F(ab')2 and Fab anti-PD-1 inhibited Treg TEM (Fig. 2c), showing that blockade was not FcR dependent or due to cross-linking of PD-1. PD-1$^{-/-}$ iTregs had reduced migration and motility compared to wild-type (Fig. 2d, e), while PD-L1$^{-/-}$ iTregs had enhanced motility (Fig. 2e), suggesting Treg PD-L1 may constitutively engage with PD-1 in cis to prevent PD-1 from interacting with LEC PD-L1. These data show that both genetic and pharmacologic inhibition resulted in similar effects. Importantly, PD-1 blockade of activated human naïve Tregs but not Teffs with anti-human PD-1 mAb (EH12.2H7) also inhibited TEM across human LECs (Fig. 2f), demonstrating similarities between murine and human migration mechanisms. Anti-PD-1 mAbs also inhibited migration to another T cell chemoattractant, sphingosine 1-phosphate (S1P) (Supplementary Fig. 3e), showing that TEM blockade was not limited to one chemokine.

CD80, which was more elevated on Teffs, is another ligand for PD-L1. To test whether these cells used the CD80/PD-L1 for TEM, we pretreated murine Teffs or iTregs with anti-CD80 (1G10) to block CD80/PD-L1. Masking CD80 inhibited Teff but not iTreg TEM (Fig. 2g) in a dose-dependent manner (Fig. 2h). Stimulation of naive CD8 T cells and B cells also increased PD-1 expression (Supplementary Fig. 2a, b), however, blocking PD-1, CD80, or PD-L1 on activated CD8 T cells or B cells did not affect cell migration (Supplementary Fig. 3f, g). Similarly, these mAbs did not block TEM of matured BMDCs (Supplementary Fig. 3h), suggesting PD-1/PD-L1 signaling did not favor CD8 T cell, B cell, or BMDC TEM, but was used uniquely by Tregs for TEM.

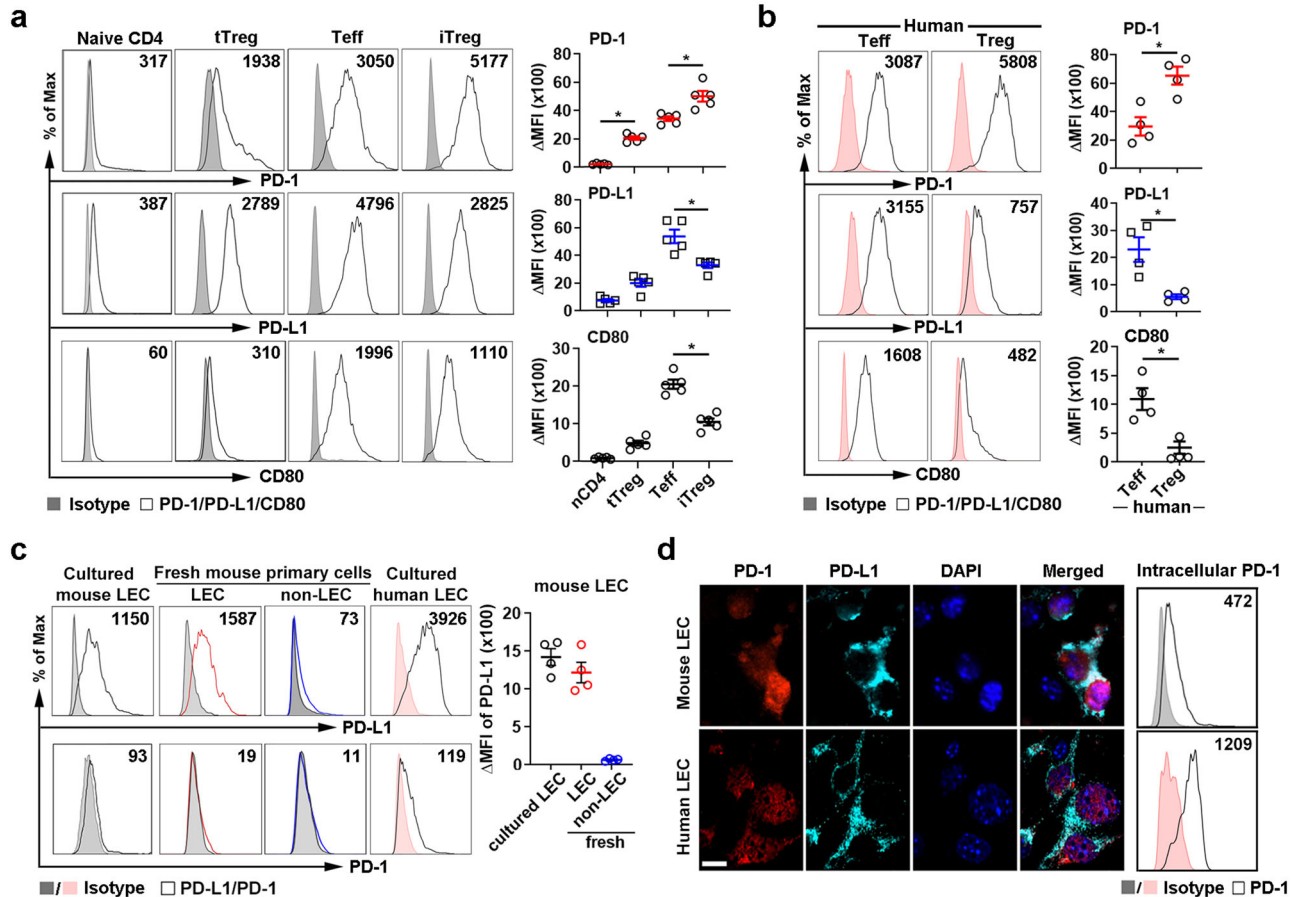

**Fig. 1 Cell surface expression of PD-1, PD-L1, and CD80 in T cell subsets and LEC. a–c** Flow cytometry analysis of PD-1, PD-L1, and CD80 expression on FACS-sorted mouse Foxp3GFP⁻CD44^lowCD25⁻CD4⁺ naïve CD4 or IL-2 + anti-CD3/28-stimulated activated Foxp3GFP⁻ CD25⁺CD4⁺ Teffs, Foxp3GFP⁺CD44^lowCD25⁺CD4⁺ tTregs or IL-2 + anti-CD3/28 + TGF-β1-induced Foxp3GFP⁺CD44^highCD25⁺CD4⁺ iTregs (**a**); human CD4⁺CD25^highCD127⁻CD45RA⁻ Tregs or CD4⁺CD25⁻CD127⁺CD45RA⁺ Teffs (**b**); or cultured or freshly isolated mouse LECs (Lyve-1⁺CD31⁺), mouse non-LECs (Lyve-1⁻CD31⁻) and cultured human LECs (**c**). ΔMFI, MFI of Ab staining minus isotype control staining. Mean ± SEM. *p < 0.001 by one-way ANOVA with Tukey's multiple comparison test (**a**, **c**) and unpaired, two-tailed t-test with Welch's correction (**b**). **d** Immunohistochemistry of PD-1 and PD-L1 expression and flow cytometry analysis of permeabilized LECs. Magnification ×60, scale bar 10 μm. Intracellular staining of PD-1 in LEC shown. Data representative of 3 (**a–c**) or 2 (**d**) independent experiments. Source data are provided as a Source Data file.

**Differential signaling in iTregs and Teffs by PD-L1 Ig engagement**. We investigated signaling in T cells by PD-L1 engagement during murine iTreg or Teff migration. Mouse PD-L1 Ig (Immunoglobulin) (extracellular domain fused with human IgG1) was immobilized on the Boyden chamber membrane. Migration of iTregs but not Teffs across the PD-L1 Ig-coated (LEC-free) membrane toward CCL19 or S1P (Supplementary Fig. 3d) was significantly enhanced (Fig. 2i) compared to membranes coated with human IgG1, CD80 Ig or LTβR Ig as controls. Engagement by PD-L1 Ig induced rapid phosphorylation of Akt (also known as protein kinase B) on threonine 308 (Thr308) in iTregs after 10 min. PD-L1 Ig suppressed classical NFκB-p65 phosphorylation, while Akt phosphorylation was maintained (Fig. 2j). Only low and transient phosphorylation of extracellular signal-regulated kinase (ERK) was observed. In contrast to iTregs, Teffs showed strong constitutive phosphorylation of ERK (Fig. 2k) which was suppressed by PD-L1 Ig. PD-L1 Ig engagement on Teffs had no specific effect on NFκB or Akt phosphorylation. These data thus indicated different responses of iTregs versus Teffs after PD-L1 engagement. PD-1 blocking mAb (Rmp1-14) pretreated iTregs and CD80 blocking mAb (1G10) pretreated Teffs did not demonstrate specific NFκB or ERK activation, while phosphorylation of Akt (Thr308) was observed (Supplementary Fig. 3i, j). These data suggested that the steric

effects of the PD-1 or CD80 blocking mAbs on the Tregs or Teffs, respectively, were most important in blocking TEM rather than directly inducing signals that impaired TEM of either T cell subset.

**Differential PD-L1 signaling on LEC regulates Treg and Teff TEM**. Since both mouse and human LECs expressed high levels of PD-L1, we investigated whether blocking LEC PD-L1 would affect TEM. Blocking LECs with anti-mouse PD-L1 mAb (10 F.9G2), which blocks both PD-1/PD-L1 and CD80/PD-L1 interactions[28], inhibited both iTreg and Teff TEM (Fig. 3a) Blocking LECs with anti-mouse PD-L1 mAb (10 F.2H11), which blocks only the CD80/PD-L1 interaction[28,29], inhibited Teff but not iTreg TEM (Fig. 3a). Anti-mouse PD-1 mAb (Rmp1-14) pretreatment of LECs had no effect on TEM for either cell type. Treating LECs with F(ab')2 or Fab anti-PD-L1 (10 F.9G2) mAbs also inhibited iTreg TEM (Fig. 3b), indicating blockade was not due to the FcR ligation or crosslinking of PD-L1. Similarly, blocking PD-L1 on human LECs with anti-human PD-L1 mAb (29E.2A3), which blocks both PD-1/PD-L1 and CD80/PD-L1 interactions, inhibited human activated Treg and Teff TEM. Blocking LEC PD-L1 with anti-human PD-L1 mAb (MIH3), which blocks only the PD-1/PD-L1 interaction[30], inhibited Treg

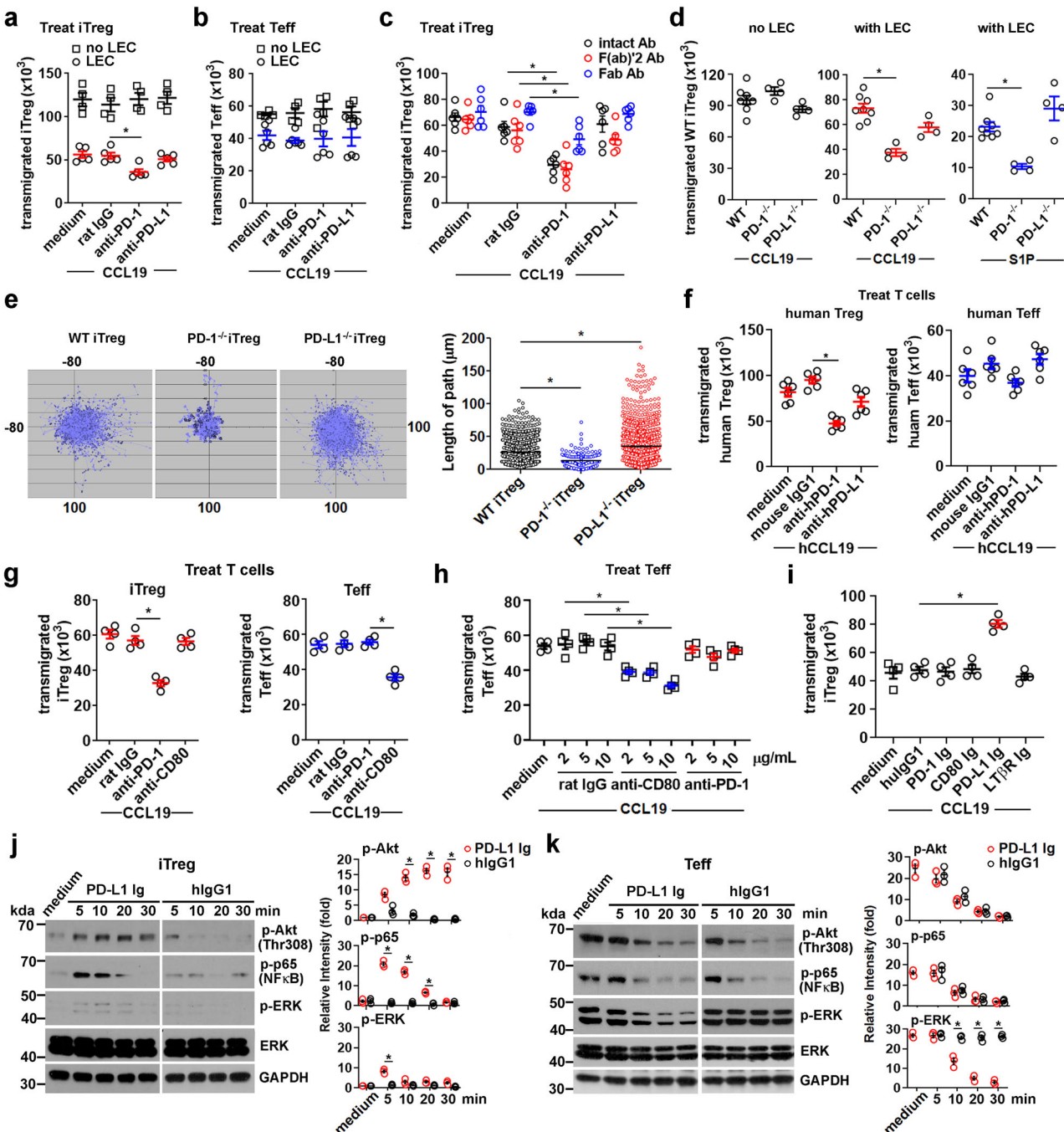

**Fig. 2 Essential roles of PD-1 and CD80 for T cell TEM. a** In vitro migration assay of iTregs treated with anti-mouse PD-1 (Rmp1-14), anti-mouse PD-L1 (10 F.9G2), or rat IgG (2A3), loaded in Boyden chambers with or without LEC. **b** Same as in **a**, in vitro migration of blocking mAb-treated Teffs. **c** Comparison of blocking intact mAb (2 μg/mL) and F(ab)'2 mAb (1 μg/mL) treatment of iTregs for TEM. **d** WT, PD-1-deficient (PD-1$^{-/-}$), and PD-L1-deficient (PD-L1$^{-/-}$) iTregs TEM. **e** Time-lapse imaging of WT, PD-1$^{-/-}$, PD-L1$^{-/-}$ iTreg movements during TEM. **f** In vitro migration of human Tregs or Teffs blocked with 5 μg/mL anti-human PD-1 (EH12.2H7), anti-human PD-L1 (29E.2A3), or mouse IgG1. **g** iTregs as in **a** or Teffs as in **b** treated with 2 μg/mL anti-mouse PD-1 (Rmp1-14), anti-mouse CD80 (1G1), or rat IgG (2A3). **h** Teffs as in **b** treat with various doses of anti-CD80 (1G1) or anti-PD-1 (Rmp1-14). **i** iTregs as in **a** loaded in Boyden chambers coated with 1 μg/mL mouse PD-L1 Ig, CD80 Ig, LTβR Ig, or mouse IgG1. **j, k** Immunoblotting for Akt, NFκB-p65, and ERK phosphorylation after PD-L1 ligation of iTregs (**j**) and Teffs (**k**) stimulated for the indicated times with 1 μg/mL mouse PD-L1 Ig, CD80 Ig, or IgG1 coated on wells. Relative band intensities shown. Data representative of 3 independent experiments (**a–k**). Mean ± SEM. *$p < 0.01$ by one-way ANOVA with Sidak's multiple comparisons test in the same group (**a–d, f–i**), $p < 0.0001$ by unpaired $t$-test (**e**), or $p < 0.001$ by unpaired, two-tailed $t$-test with Welch's correction (**j, k**). Source data are provided as a Source Data file.

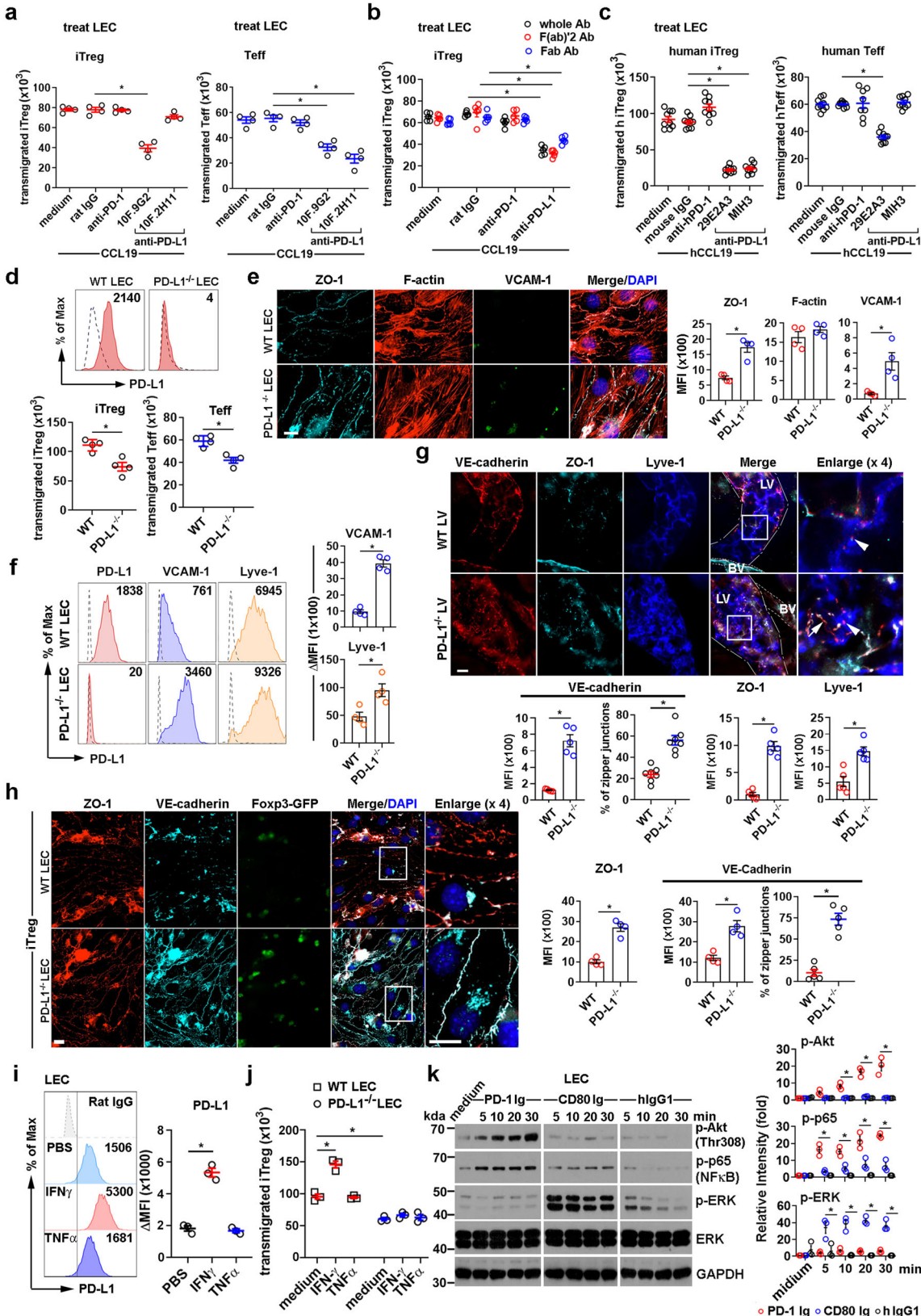

but not Teff TEM. Blocking PD-1 on LECs with anti-human PD-1 (EH12.2H7) had no effect on TEM (Fig. 3c). The TEM of Tregs and Teffs across CRISPR/Cas9-knockout PD-L1$^{-/-}$ LECs was also inhibited (Fig. 3d). Interestingly, resting CRISPR/Cas9-knockout PD-L1$^{-/-}$ LECs expressed higher levels of zonulin-1 (ZO)-1, VCAM-1, and Lyve-1 compared to WT LECs, while

F-actin remained unchanged (Fig. 3e-f; Supplementary Fig. 4a). Notably, lymphatic vessels (LV) of PD-L1$^{-/-}$ mice had increased zipper junctional VE-cadherin and button junctional ZO-1 (Fig. 3g), supporting the functional importance of the organization of the junctional proteins in endothelial buttons and zippers[31,32]. PD-L1$^{-/-}$ LVs also had increased VCAM-1 and

**Fig. 3 Differential PD-L1 signaling on LEC regulates Treg and Teff TEM. a–d** TEM assays. LEC treated with 2 µg/ml intact (**a**), F(ab)'2 or Fab (**b**) anti-mouse PD-1 (Rmp1-14), or with 5 µg/ml anti-mouse PD-L1 (10 F.9G2), anti-mouse PD-L1 (10 F.2H11), or rat IgG (2A3), loaded with iTregs or Teffs (as in Fig. 2). Human LECs (**c**) treated with 5 µg/mL anti-human PD-1 mAb (EH12.2H7), anti-human PD-L1 mAb (29E.2A3), or mouse IgG1, then loaded with activated human iTregs or Teffs (**d**) Flow cytometry for PD-L1 on iTreg or Teff migration across wild-type (WT) or PD-L1$^{-/-}$ LEC. **e** Immunohistochemistry for ZO-1, F-actin, and VCAM-1 expression in resting WT or PD-L1$^{-/-}$ LECs. **f** Flow cytometry for PD-L1, VCAM-1, and Lyve-1 in resting WT or PD-L1$^{-/-}$ LECs. **g** Whole-mount ear staining for VE-cadherin, ZO-1, and Lyve-1 in WT and PD-L1$^{-/-}$ mice. LV lymphatic vessel, BV blood vessel. Arrow heads indicate button junctions; arrows indicate zipper junctions. **h** Immunohistochemistry for ZO-1 and VE-cadherin in WT or PD-L1$^{-/-}$ LECs after iTreg TEM (LECs from same experiment as panel **d**). Quantification of zipper junctions in LVs or LECs. Each dot represents one LV or every 10 LECs (**g**, **h**). **i** Flow cytometry analysis of PD-L1 expression on LEC stimulated with 100 ng/mL mouse IFNγ or 20 ng/mL mouse TNFα for 16 h at 370 °C. Representative histograms shown. **j** iTregs loaded to Boyden chamber with LECs treated as in **i**. **k** Immunoblots for Akt, NFκB-p65, and ERK phosphorylation in LECs stimulated with 1 µg/mL mouse PD-1 Ig, mouse CD80 Ig, or human IgG1 for the indicated times. Relative band intensities shown. Data representative of 3 independent experiments (**a–k**). Magnification ×60 (**e**, **g**, **h**); scale bar 42 µm (**e**, **h**). 14 µm (**g**), MFI shown (**e–i**). Mean ± SEM (**a–k**). *p < 0.01 by one-way ANOVA with Sidak's multiple comparisons test (**a–c**, **i**, **j**); *p < 0.01 versus WT (**d–h**) or hIgG1 (**k**) by unpaired, two-tailed t-test with Welch's correction. Source data are provided as a Source Data file.

Lyve-1 expression (Supplementary Fig. 4b). Notably, CRISPR/Cas9-knockout PD-L1$^{-/-}$ LECs exhibited a broad spindle shape with F-actin extending to both ends of the cell, unlike WT LECs which had more compact morphology with irregular actin filaments surrounding the nuclei (Fig. 3e, Supplementary Fig. 4c). Further phenotypic characterization showed that the VE-cadherin expression remained high at zipper junctions after Tregs migrated across PD-L1$^{-/-}$ LECs, while in WT LECs VE-cadherin was only sporadically located in button junctions (Fig. 3h). These observations suggested specific PD-L1 regulated structures on LECs, along with PD-1/PD-L1 signals between Tregs and LECs, modulate intercellular junctions for TEM. The previous reports[22,33,34] demonstrated that PD-L1 protects LN LECs and tumor cells from apoptosis, while we observed that loss of PD-L1 in dermal LEC increased cell viability and prevented apoptosis induced by IFNγ but not TNFα. Notably, CD80 Ig also decreased LEC viability and was PD-L1 dependent (Supplementary Fig. 4e -f). PD-L1 expression was enhanced by IFNγ but not TNFα, and IFNγ treatment of LEC promoted iTreg TEM (Fig. 3i). No specific signaling was induced in LECs by anti-PD-L1, since both blocking Ab (10F9G2) and isotype control rat IgG (2A3) only induced transient nonspecific phosphorylation of ERK (Supplementary Fig. 4g). These data again suggested that the steric effects of the mAbs on the LECs were most important in blocking TEM rather than directly inducing signals that impaired TEM.

**LEC PD-L1 signals through NFκB-p65, ERK, and PI3K/Akt to regulate the endothelial structure and enhance TEM.** We next investigated PD-L1 signaling in LECs and whether it regulated the expression and structure of the migration molecules VCAM-1 and VE-cadherin. LEC PD-L1 signaling was activated by ligation with PD-1 Ig or CD80 Ig. PD-1 Ig induced phosphorylation of classical NFκB (p65), Akt (Thr308), and ERK (Fig. 3j). In contrast, CD80 Ig engagement induced strong phosphorylation of ERK and modest classical NFκB activation. CD80/PD-L1 did not activate Akt (Thr308) (Fig. 3j). PD-1 Ig augmented VCAM-1 expression (Fig. 4a, upper panel), which was inhibited by blocking the classical NFκB pathway with BAY11-7082, but not by blocking ERK with U1062 or PI3K/Akt with PI3Kin. In contrast, blocking PI3K/Akt signaling lead to enhanced VCAM-1 expression (Fig. 4a, middle panel). CD80 Ig also enhanced VCAM-1 expression (Fig. 4a, upper panel), which was inhibited by blocking both classical NFκB and ERK (Fig. 4a, bottom panel). These data indicated that LEC VCAM-1 expression was directly regulated by both NFκB p65 and ERK.

PD-1 Ig induced PD-L1 signaling also decreased junctional VE-cadherin, decreased zipper junctions, and increased button-like junctions. Interestingly, VE-cadherin expression was not affected by CD80 Ig (Fig. 4b, upper panel). Decreased VE-cadherin zipper junction was restored by blocking PI3K/Akt and ERK but not NFκB-p65 (Fig. 4b, middle panel), suggesting PD-1/PD-L1 signals through PI3K/Akt (Thr308) and ERK to modulate junctional VE-cadherin. Similar increases in VCAM-1 and downregulation of junctional VE-cadherin with more button-like junctions were observed when PD-1$^{high}$ iTregs migrated across LEC layers (Fig. 4c, bottom left panel). In contrast, Teffs and PD-1$^{-/-}$ iTregs did not cause these changes to cell adhesion and junctional molecules (Fig. 4c, right panels). These results demonstrated that Tregs stimulated outside-in signaling to LECs via PD-L1

**Migrated iTregs but not Teffs retain PD-1 expression.** We investigated the changes in cell surface PD-1 or PD-L1 expression in iTregs, Teffs, and LECs before and after their migration across LECs. Migrated iTregs maintained higher PD-1 expression than non-migrated iTregs. In contrast, migrated Teffs had lower PD-1 expression than non-migrated (Fig. 5a, lower panel). PD-1 on iTregs or Teffs remain unchanged after crossing LEC-free Boyden chamber membranes (Fig. 5a upper panel). Similarly, migrated activated human naïve Tregs but not CD4 Teffs retained higher PD-1 expression than non-migrated after crossing human LECs (Fig. 5b). Notably, PD-L1 expression on the mouse and human LECs was decreased after Treg but not Teff migration (Fig. 5c), suggesting PD-1/PD-L1 signaling induced LEC PD-L1 internalization or degradation.

**Treg PD-1 or Teff CD80 trans-binds PD-L1 on LEC for TEM.** Confocal analysis of PD-1/PD-L1 interactions revealed that these molecules colocalized on the interface between Tregs and LECs (Fig. 5d, arrowhead in middle panel), but not between Teffs and LECs, suggesting the direct interaction between Treg PD-1 and LEC PD-L1 during TEM. In contrast, CD80 but not PD-1 on Teffs colocalized with WT LEC PD-L1 (Fig. 5e, arrowhead in the bottom left panel), and CD80 accumulated in the interface between Teff and WT LEC, but not Teff and PD-L1$^{-/-}$ LEC (Fig. 5e. bottom right panel). CD80 in APCs reportedly binds PD-L1 only in cis and prevents engagement of T cell PD-1 to enhance immunity[35,36]. Given CD80 to PD-L1 signaling in LEC noted above, we hypothesized that CD80 also binds in trans to LEC PD-L1. Since LECs have no surface CD80 or PD-1 expression, by incubating LECs with CD80 Ig or other receptors fused with human IgG1, we observed binding of CD80 Ig or PD-1 Ig but not CTLA4 Ig to LEC PD-L1, using immunohistochemistry (Fig. 5f, g), flow cytometry (Fig. 5h), and co-immunoprecipitation (Fig. 5i). Teffs also bound to immobilized PD-L1 Ig, and binding was blocked by masking PD-L1 on PD-L1 Ig; or CD80 on Teffs (Fig. 5j left). Further, PD-L1 Ig coated

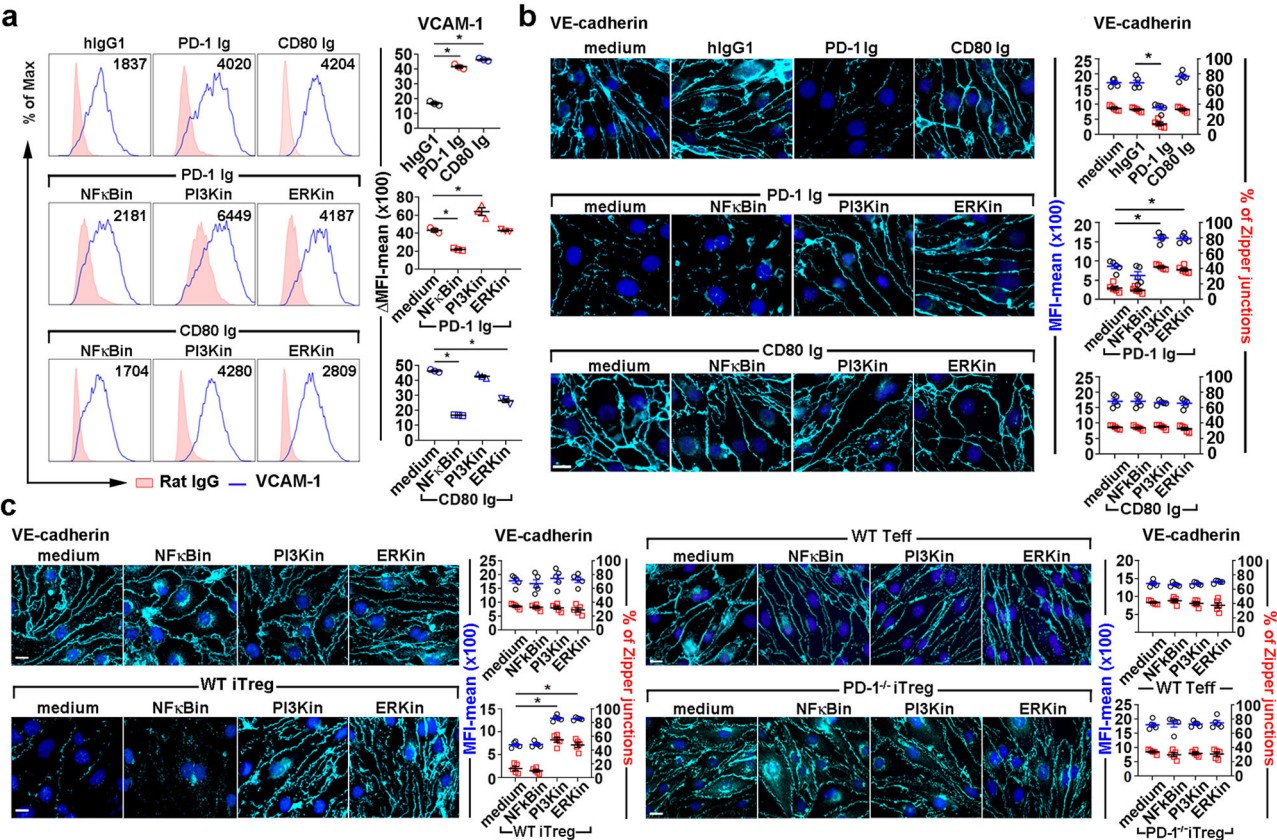

**Fig. 4 PD-L1 signals through NFκB-p65, ERK, and PI3K/Akt pathways to regulate endothelial structure. a–c** LECs pretreated with classical NFκB inhibitor (NFκBin, 5 μM), PI3K inhibitor (PI3Kin, 0.5 μM), or ERK inhibitor (ERKin, 5 μM) for 30 min at 37 °C, then stimulated with soluble 1 μg/mL PD-1 Ig, CD80 Ig, or incubated with $2 \times 10^5$ WT or PD-1$^{-/-}$ iTregs or WT Teffs for 16 hours at 37 °C. VCAM-1 surface expression analyzed by flow cytometry (**a**). Intercellular VE-Cadherin analyzed by immunohistochemistry (**b**, **c**). Quantification of zipper junctions in LECs. Each dot represents every 10 LECs. Magnification ×60, scale bar 10 μm (**b**) and 20 μm (**c**). Data representative of 3 independent experiments. (**a**–**c**) Mean ± SEM. *$p < 0.01$ by one-way ANOVA with Sidak's multiple comparisons test. Source data are provided as a Source Data file.

in the Boyden chamber increased Teff TEM, which was blocked by anti-CD80 (1G10). Masking the PD-L1 Ig with anti-PD-L1 (10 F.9G2) also blocked Teff TEM (Fig. 5j right). Incubating Teffs with increasing doses of CD80 Ig did not alter PD-L1 expression (Supplementary Fig. 4h), suggesting that there was no substantial CD80/PD-L1 cis binding on Teffs. Additionally, CD80 Ig did not bind to PD-L1$^{-/-}$ LEC (Fig. 5g). Taken together, the data suggest Teffs use CD80 to engage LEC PD-L1 during TEM. Consistent with that idea, PD-1 but not CD80 stimulated predominantly LEC NFκB p65 activation (Fig. 3k), and PD-1$^{high}$ iTregs also rapidly induced NFκB p65 nuclear translocation in LEC (Fig. 5k, arrow in middle panel). In contrast, LECs that interacted with Teffs had no NFκB p65 nuclear translocation (Fig. 5k, bottom panel).

**PD-1/PD-L1 and CD80/PD-L1 interactions regulate iTreg and Teff afferent lymphatic migration, respectively.** The role of PD-1/PD-L1 for in vivo iTreg afferent lymphatic migration was assessed by the footpad migration assay. WT iTregs or Teffs were pretreated with anti-PD-1 (Rmp1-14), anti-PD-L1 (10 F.9G2), anti-CD80 (1G10), or isotype control rat IgG2a prior to injection into hind footpads. Popliteal draining LNs (dLNs) were collected 16 h later and migrated cells enumerated by flow cytometry. Anti-PD-1 blocked iTreg migration to the dLNs (Fig. 6a left), while migration of Teffs was inhibited by anti-CD80 blockade but not by PD-1 blockade (Fig. 6a right). Similarly, intradermal injection of footpads with dual blocking anti-PD-L1 mAb (10F9G2) prior to transferring the T cells inhibited Treg (Fig. 6b left) and Teff

(Fig. 6b right) migration to dLNs. Additionally, PD-1$^{-/-}$ iTregs also showed impaired migration into the dLNs (Fig. 6c).

**Frequencies of CD25$^-$Tregs and CD25$^+$Teffs are regulated in TILs and dLNs of melanoma by PD-1 and CD80 blockade, respectively.** To assess the role of PD-1/PD-L1 and CD80/PD-L1 for in vivo iTreg and Teff recruitment into tumor-infiltrating lymphocytes (TILs) and tumor dLNs, we treated B16F10 melanoma bearing mice with anti-PD-1 (Rmp1-14) or anti-CD80 (1G10) mAbs that blocked their specific binding to PD-L1, and assessed tumor growth and changes in Tregs and Teffs in TILs and dLNs (Fig. 6d). Anti-PD-1 mAbs inhibited melanoma tumor growth (Fig. 6e), consistent with prior reports[37]. Anti-CD80 mAbs also inhibited tumor growth (Fig. 6e). CD25$^-$Foxp3$^+$CD4 Tregs significantly increased in melanoma TILs and decreased in dLNs after treatment with anti-PD-1 but not anti-CD80 mAbs, implying PD-1 blockade prevented CD25$^-$Foxp3$^+$CD4 egress from tumor to dLNs. It also suggested that migration inhibition may have resulted in the conversion of suppressive CD25$^+$ Tregs to non-suppressive or effector-like CD25$^-$ Tregs[38]. In contrast, CD80 but not PD-1 blockade caused Foxp3$^-$CD25$^+$CD4 Teffs to significantly increase in TILs and decrease in dLNs (Fig. 6f, g, Fig. 7a), indicating CD80 blockade may specifically regulate CD25$^+$CD4 Teff lymphatic egress of TILs into dLNs. The frequency of proliferating Ki67$^+$ CD4 iTregs in TILs remained unchanged after PD-1 or CD80 blockade; however, Ki67$^+$ CD4 Teffs or non-Tregs were remarkably increased in TILs

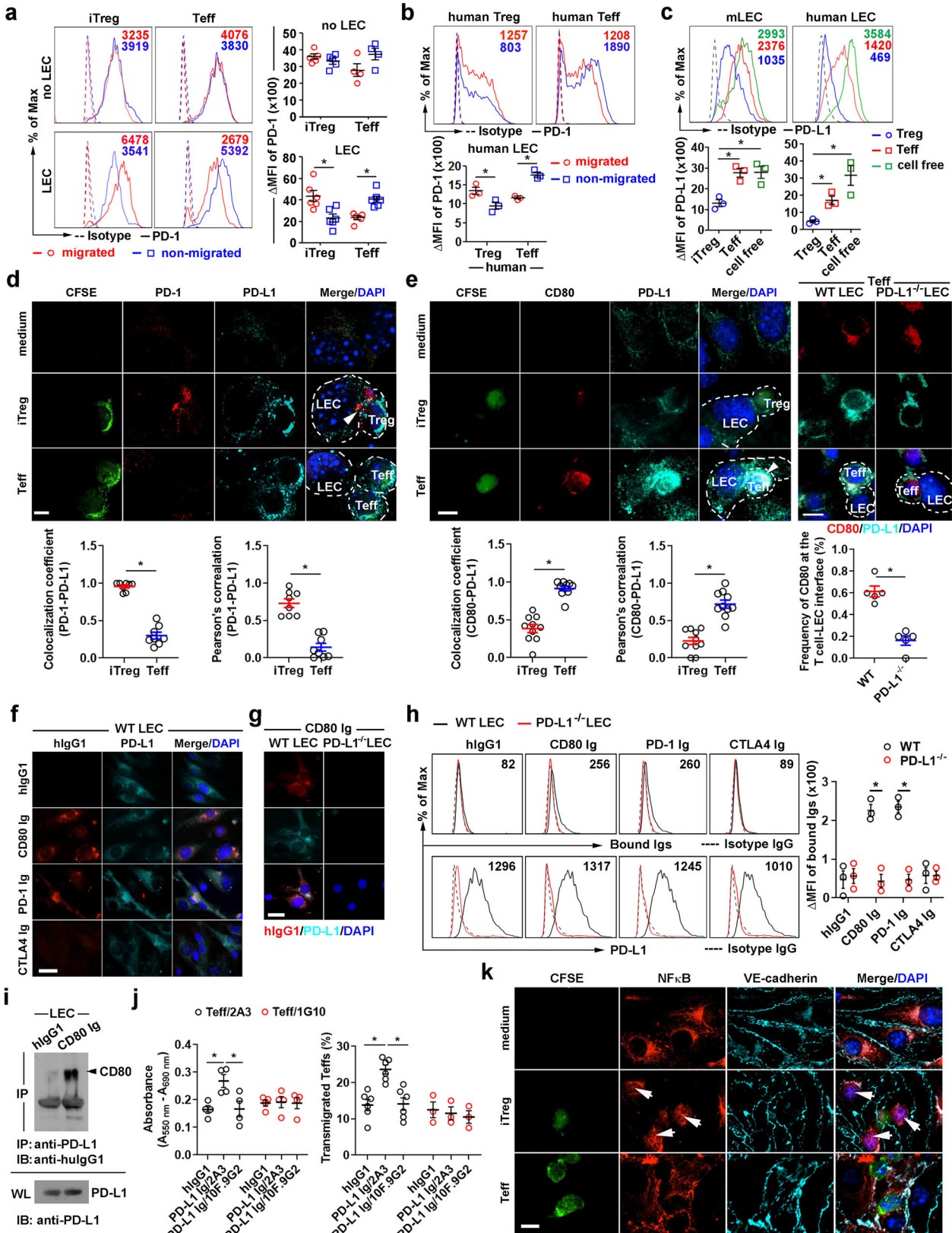

(Supplementary Fig. 5a-d), suggesting that PD-1 or CD80 blockade did not alter Treg proliferation but did promote CD4 Teff or non-Treg proliferation. In dLNs, there was no alteration of Treg proliferation, while non-Treg CD4 T cells had increased proliferation after PD-1 blockade (Fig. 7a, Supplementary Fig. 5e, f). PD-1 blockade increased tumor infiltration by both CD4 and CD8

T cells (Supplementary Fig. 6a), reflecting the proliferation of non-Tregs due to reduced suppression from CD25⁻ Tregs. PD-1 blockade reinvigorated the exhausted T cells by reducing the fractions of T cell immunoglobulin and mucin domain-3 (TIM-3)⁺ and Lymphocyte-Activation Gene 3 (LAG-3)⁺ CD4 TILs or TIM-3⁺ CD8 TILs (Supplementary Fig. 5b, c), and

**Fig. 5 Differential crosstalk between migrating iTregs or Teffs and LECs. a** Flow cytometry analysis of PD-1 on iTregs or Teffs after 3 h of TEM across mouse LECs. Values are the ΔMFIs for the migrated and non-migrated populations. **b, c** Flow cytometry analysis of PD-1 on human Tregs or Teffs (**b**) or PD-L1 on mouse or human LECs (**c**) after 3 h of TEM across LECs. **d, e** Analysis of interaction of PD-L1 with PD-1 (**d**), or with CD80 (**e**) on the interface (arrowhead) of CFSE-labeled iTregs or Teffs and LECs after 3 h of TEM. Colocalization coefficient and Pearson's correlation of T cell PD-1 or CD80 interaction with LEC PD-L1 shown. Each dot represents the region of interest (ROI) of the interface. Percentage of CD80 at the T-cell-WT or PD-L1$^{-/-}$LEC interface also shown. **f–i** Analysis of CD80 Ig binding to PD-L1 on LEC by immunohistochemistry (**f, g**), flow cytometry (**h**), and immunoprecipitation (**i**). **j** Teffs bind immobilized PD-L1 Ig (left panel) which promote TEM (right panel). **k** Analysis of NFκB-p65 activation in LEC 30 min after incubation with iTregs or Teffs. Arrows indicate the nuclear translocated NFκB-p65. Magnification ×60 (**d–g, k**), scale bar 5 μm (**d**), 7 μm (**e, k**), and 14 μm (**f, g**). Data representative of 3 independent experiments. Mean ± SEM (**a–c, h, j**). *p < 0.01 by one-way ANOVA with Sidak's multiple comparisons test (**c, j**) and unpaired, two-tailed t-test with Welch's correction (**a, b, h**). Source data are provided as a Source Data file.

enhancing the granzyme B and IFNγ-producing CD4 or CD8 TILs (Supplementary Fig. 5d, e). CD80 blockade resulted in almost identical effects, except that IFNγ-producing CD8 remained unchanged (Supplementary Fig. 5e). LAG-3 expression on CD8 TILs was not affected by either blockade (Supplementary Fig. 5c).

**CD25$^-$Foxp3$^+$ CD4 Tregs in TILs resemble "fragile Tregs".**
CD25$^-$Foxp3$^+$ but not CD25$^+$Foxp3$^+$ CD4 Tregs in TILs expressed high levels of IFNγ and the Th1 transcription factor T-bet, as well as higher PD-1 expression (Fig. 7b), resembling the "fragile Tregs" defined by the retention of Foxp3 expression with loss of suppressive function[39]. This intratumoral phenotype was less pronounced in dLNs (Fig. 7b). After PD-1 blockade, the percentage of IFNγ$^+$T-bet$^+$PD-1$^+$CD25$^-$Tregs in TIL CD4 T cells was increased (Fig. 7c). In contrast, this population decreased in dLN CD4 T cells (Fig. 7c), consistent with PD-1 blockade preventing CD25$^-$Foxp3$^+$Treg egress from the tumors into dLNs (Fig. 6f, g). CD25$^+$FoxP3$^+$ CD4 Tregs expressed lower levels of IFNγ, T-bet, and PD-1 (Fig. 7b). After PD-1 or CD80 blockade, the frequencies of IFNγ, T-bet, and PD-1 -expressing CD25$^+$ Tregs did not change in CD4 cells in either TILs or dLNs (Fig. 7c). CD80 but not PD-1 blockade increased IFNγ- and T-bet- expressing CD4 Teffs in TILs, while PD-1-expression on CD4 Teffs or CD25$^+$ Tregs in TILS was not altered by PD-1 or CD80 blockade (Fig. 7c, d). We next sought to determine the direct effect of PD-1 blockade on Treg migration and conversion in melanomas. CD25$^+$Foxp3GFP$^+$CD4$^+$ iTregs from Foxp3GFP C57BL/6 (CD45.1) mice were isolated and pretreated with anti-PD-1 mAb (Rmp1-14) or isotype control rat IgG2a (2A3). These cells were then transferred intratumorally to B16F10-bearing CD45.2 C57/BL6 hosts 8 days after tumor inoculation. Sixteen hours later, the tumors and dLNs were analyzed (Fig. 7e). Transferred iTregs with PD-1 blockade had reduced migration to the dLNs and were retained in the TILs (Fig. 7f). Notably, the majority of the retained Tregs in TILs after PD-1 blockade were CD25 negative with higher IFNγ and T-bet expression compared to the isotype-treated cells (Fig. 7g), suggesting parallel events of impaired Treg migration and Treg conversion by PD-1 blockade.

## Discussion

In the present study, we demonstrated that Tregs and especially activated iTregs, preferentially expressed PD-1 to ligate PD-L1 for lymphatic transendothelial migration (TEM). The direct role in migration and the preferential use by iTreg but not non-Treg also suggests that PD-1/PD-L1 signaling may be important to modulate the ratio of suppressive Tregs and reactive Teffs at inflammatory sites. PD-1 signaling in Teffs recruits SHP2 to terminate Zap70/ERK and PI3K/PKCθ and counteracts T cell receptor signal transduction and CD28 co-stimulation[16]. In contrast in activated iTregs we found that PD-1 signaling suppressed classical NFκB-p65 and induced upregulation of ERK and Akt

phosphorylation, indicating alternative PD-1 signaling compared to Teffs.

iTregs expressed higher levels of PD-1 than Teffs, while Teffs expressed higher levels of PD-L1 and CD80 than Tregs. These differences were more pronounced for human Tregs and Teffs. In earlier studies[30,40], CD80 reportedly trans-binds PD-L1 and inhibits immune responses. More recent studies suggest CD80 binds exclusively in cis to PD-L1 on APC[35,36], promoting immunity. These differences might be due to variable spatio-temporal dynamics of surface expression and different cell types with unique functions. Differing from studies that relied on engineered tumor cell lines with gene overexpression in B cells or APCs, our model exclusively used primary LECs, which do not express CD80 or PD-1, to show a trans interaction of CD80/PD-L1. In particular, we demonstrated that: (i) CD80 Ig induced strong PD-L1-ERK signaling in wild-type but not PD-L1$^{-/-}$ LECs which was non-overlapping with PD-1 Ig induced PD-L1-PI3K/AKT signaling; (ii) masking CD80 on Teffs or blocking PD-L1 on LEC with specific PD-L1 antibody (10 F.2H11), which solely blocks the PD-L1 and CD80 interaction, inhibited exclusively Teff TEM; (iii) protein binding assays indicated CD80 Ig and PD-1 Ig but not CTLA4 Ig bound to LEC PD-L1; and LEC PD-L1 co-immunoprecipitated with CD80 Ig; (iv) Teffs bound to immobilized PD-L1 Ig which was blocked by masking PD-L1 on PD-L1 Ig or CD80 on Teff. Further, the PD-L1 Ig coated on the Boyden chamber increased Teff TEM which was blocked by anti-CD80. Masking PD-L1 Ig with anti-PD-L1 also blocked Teff TEM. (v) incubating Teffs with increasing doses of CD80 Ig did not alter PD-L1 expression; and (vi) anti-PD-L1 treatment of Teffs did not affect TEM. Taken together, we propose the co-existence of both cis and trans binding of CD80 to PD-L1. It is possible that high levels Teff-PD-L1 occupy the lower levels of PD-1 in cis, freeing CD80 to engage LEC PD-L1 in trans for TEM (Supplementary Figs. 7, 8b). In contrast, the higher expression of PD-1 enables Tregs to engage PD-L1 on LECs and overshadows the lower expression of CD80 that precludes its participation in TEM. Other activated immune cells such as CD8 T cells, B cells, or imBMDC did not use these molecules for TEM, although they also expressed various levels of them. This highlights some of the unique attributes of Treg migration, including their ability to facilitate the migration of other cells across LEC[25].

Ligation of LEC with PD-1 Ig activated AKT phosphorylation and upregulated VCAM-1 and downregulated VE-cadherin zipper junctions. Ligation with CD80Ig only activated ERK signaling and upregulated VCAM-1, indicating different consequences of PD-1/PD-L1 versus CD80/PD-L1 engagement. Blocking LEC PD-L1 with anti-PD-L1 (10 F.9G2) which blocks interaction with both PD-1 and CD80, inhibited iTreg and Teff migration. Blocking LEC PD-L1 with anti-PD-L1 (10 F.2H11) which blocks only the PD-L1/CD80 interaction, inhibited Teff but not iTreg migration. Likewise in human LECs, masking PD-L1 with anti-PD-1 (29E.2A3) which blocks interactions with both PD-1 and

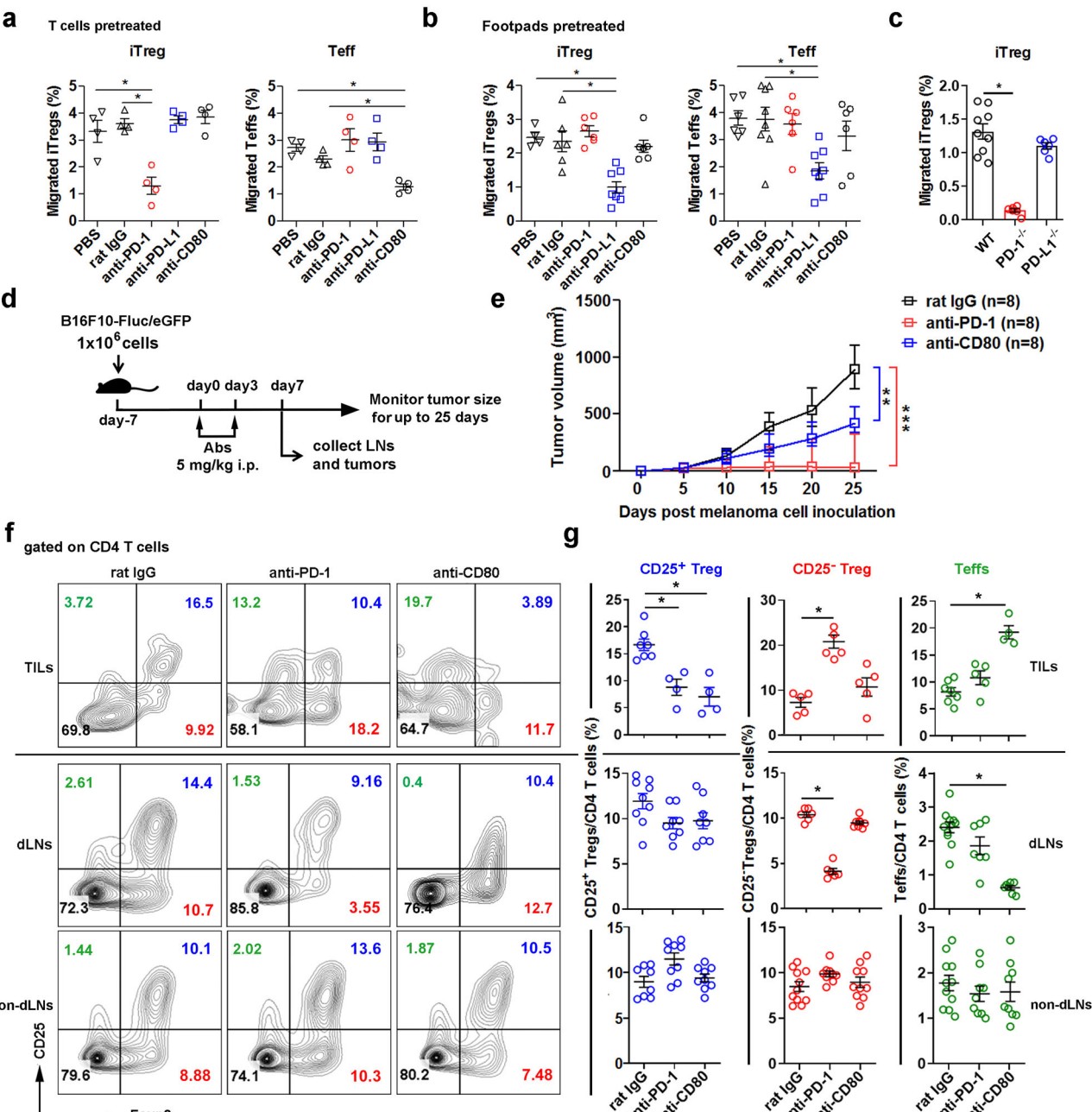

**Fig. 6 PD-1 and CD80 blockade result in CD25-Treg and Teff accumulation in TILs, reduction in dLNs, and tumor regression. a–c** In vivo footpad migration assays. **a** mAb-treated iTregs (left panel) or Teffs (right panel). **b** mAb pretreated footpads receiving iTregs (left) or Teffs (right). **c** untreated footpads receiving WT and PD-1$^{-/-}$ iTregs. $1 \times 10^6$ cells transferred, and the migrated cells (CFSE+) enumerated as the percentage of total CD4 T cells in popliteal dLNs. **d–g** B16F10-Fluc/eGFP tumor-bearing C57BL/6 mice treated with 5 mg/kg mouse weight of anti-PD-1 (Rmp1-14), anti-CD80 (1G10), or isotype rat IgG (2A3). Scheme of tumor treatment (**d**). Tumor growth curve (**e**). 8 mice/group; *$p < 0.05$, Two-way ANOVA, Holms-Šidák correction for multiple comparisons. Representative dot plots (**f**) and frequencies (**g**) of Foxp3+CD25+CD4Tregs, Foxp3+CD25−CD4 Tregs, and Foxp3−CD25+CD4 Teffs in total CD4 T cells of tumor-infiltrating lymphocytes (TILs) and of draining LNs (dLNs) or non-draining control LNs (non-dLNs) analyzed by flow cytometry. **f, g** Three to four mice/group. Data representative of 3 (**a–c, f, g**) and 2 (**e**) independent experiments. Mean ± SEM. *$P < 0.05$ by one-way ANOVA. Source data are provided as a Source Data file.

CD80, inhibited Treg and Teff, while anti-PD-L1 (MIH3) which blocks only PD-L1/PD-1 but not PD-L1/CD80, inhibited only Treg TEM. These observations strongly suggest that Tregs predominantly use PD-1/PD-L1, while Teffs use CD80/PD-L1 interactions for lymphatic TEM (supplementary Figs. 6, 7). PD-1$^{-/-}$ iTregs had decreased motility and migration on LECs, indicating PD-1 functions for T cell movements, complementing previous reports that blockade of PD-1 decreased T cell

motility[41]. Of note, PD-L1$^{-/-}$ iTregs showed increased motility and migration suggesting that Treg PD-L1 might be indirectly regulating migration in a cis fashion. Consistent with the report that PD-L1-deficiency increased adhering junction proteins in activated endothelial cells[42], Zipper junctional VE-cadherin, VCAM-1, and ZO-1, which are crucial for endothelial junctional integrity[31,32], were upregulated on PD-L1$^{-/-}$ LEC. After iTregs migrated across PD-L1$^{-/-}$ LECs, the elevated button junctional

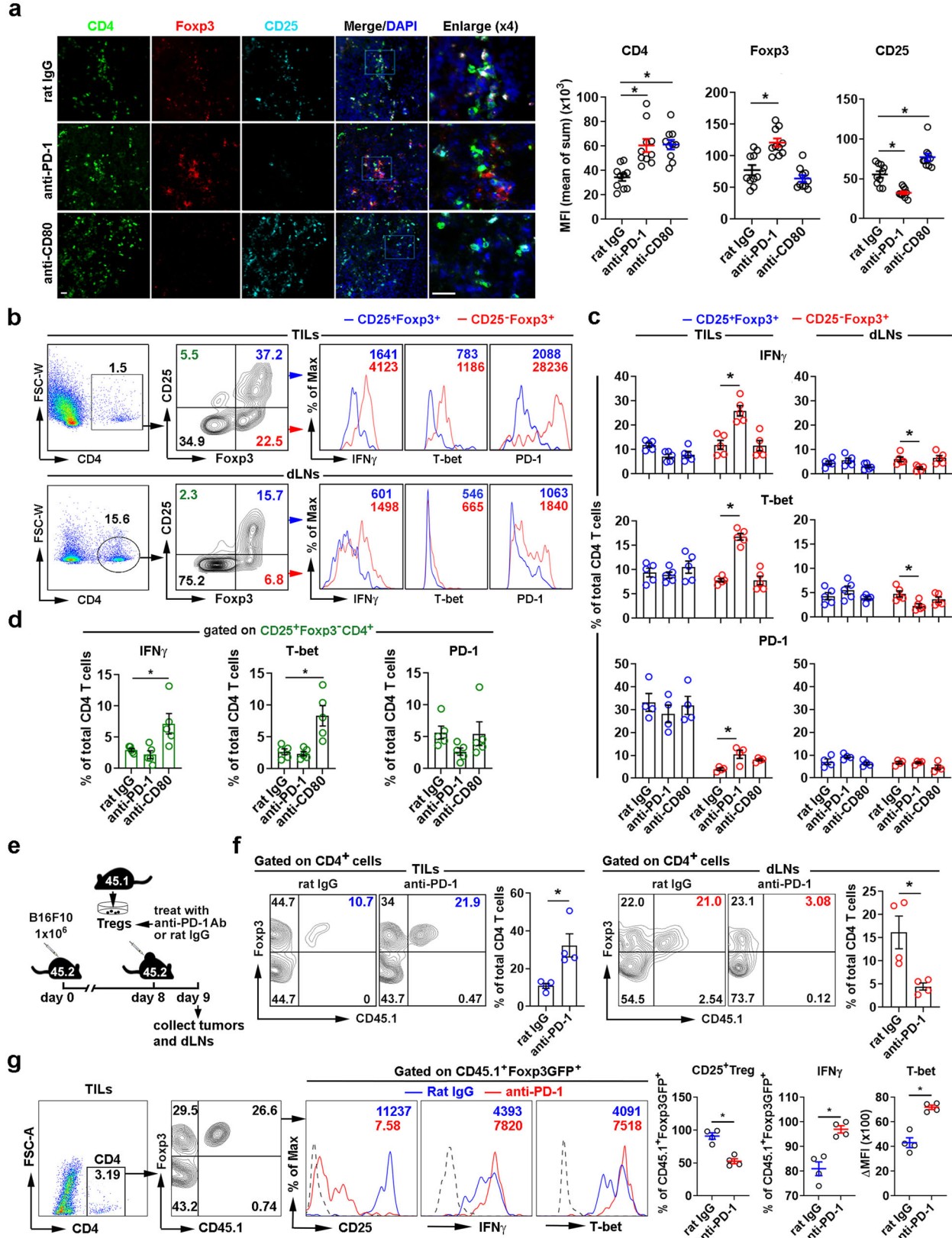

ZO-1 and the zipper junctional VE-cadherin remained. PD-L1 deficiency also upregulated Lyve-1 expression, the receptor of hyaluronan. Changes in each of these junctional and cell surface proteins could contribute to impaired T cell TEM. PD-L1$^{-/-}$ LECs and lymphatic vessels also showed altered morphology compared to WT. These observations suggested endothelial

PD-L1 is required for cytoskeletal integrity during homeostasis and morphologic changes during TEM.

PD-1 blockade with anti-PD-1 mAbs is now accepted clinical immunotherapy for melanoma[43]. The efficacy has been attributed to its reinvigoration of Teff functions. Immune suppressive Tregs in TILs are considered a barrier to effective antitumor immunity

**Fig. 7 PD-1 blockade prevents afferent lymphatic migration of CD25$^+$Foxp3$^+$CD4 Tregs from TILs to the dLNs and increases Treg conversion to CD25- IFNγ highTregs. a** Immunohistochemistry analysis of Treg subsets in TILs of mice bearing melanoma treated with anti-PD-1 (Rmp1-14), anti-CD80 (1G10), and isotype rat IgG. MFI of Foxp3, CD25, and CD4. Magnification ×20, scale bar 80 μm. **b, c** Intracellular IFNγ and T-bet, or surface PD-1 expression in CD25$^+$ or CD25$^-$ Treg subsets of TILs and dLNs, in mice treated as in (**a**) and assessed by flow cytometry. Representative gating strategy, dot plots, and histograms in TILs and dLNs (**b**) and summary of cell frequencies of all treated groups (**c**). **d** IFNγ, T-bet, and PD-1 expression in CD4 Teffs in TILs. **e–g** Transfer of anti-PD-1 treated CD25$^+$Foxp3GFP$^+$CD4$^+$ iTregs (1 × 10$^6$) into B16F10-bearing mice. Scheme of intratumoral cell transfer (**e**). Flow cytometry analysis of transferred Treg numbers (**f**) and IFNγ and T-bet expression (**g**) in TILs and dLNs. Representative dot blots and histograms shown. Data representative of 3 (**a–d**) or 2 (**e–g**) independent experiments (4 mice/group). Mean ± SEM. *$P < 0.05$ by one-way ANOVA. Source data are provided as a Source Data file.

and their depletion by anti-CD25 mAbs improves checkpoint blockade[44]. The effects of PD-1 blockade on intratumoral Tregs have been inconsistent in various clinical observations or murine models. PD-1 blockade reportedly decreased CD4 Tregs: Teffs ratios in TILs of a murine osteosarcoma model[45]. In contrast, an enhanced ratio was observed in squamous cell carcinomas[46]. In line with several prior reports[14,47], we observed that PD-1 blockade decreased CD25$^+$Foxp3$^+$ CD4 Tregs, but increased CD25$^-$Foxp3$^+$ CD4 Tregs in TILs. It is plausible that the PD-1 blockade induced conversion of CD25$^+$Foxp3$^+$ CD4 Tregs into CD25$^-$Foxp3$^+$ CD4 Tregs[38], since we observed the intratumoral transferred CD25$^+$Tregs with PD-1 blockade had decreased tumor egress, along with increased conversion to IFNγ-producing CD25-Tregs. Despite the significant increase in the CD25$^-$Treg population in the TILs, this subset was significantly decreased in the dLNs by PD-1 blockade, consistent with inhibition of tumor egress. The CD25$^-$ but not CD25$^+$ Tregs in TILs expressed high levels of PD-1, which is coincident with the in vitro migration assays showing that migrated Tregs retained high PD-1 expression, while PD-1$^{low}$ Tregs remained non-migrated. These results imply that the PD-1$^{high}$CD25$^-$Foxp3$^+$CD4 Tregs were targeted for migration inhibition by PD-1 blockade. Importantly, T-bet-associated IFNγ production was elevated in the PD-1$^{high}$CD25$^-$ Treg subset, which drives attenuated suppressive function and promotes antitumor immunity[39]. Human TIL PD-1$^{high}$Tregs are also reportedly converted to a dysfunctional signature and exhibit enhanced secretion of IFNγ after anti-PD-1 treatment[48]. It is important to note that non-mutually exclusive mechanisms for the accumulation of IFNγ-producing CD25$^-$ Foxp3$^+$CD4 Tregs in TILs due to PD-1 blockade may not only involve Treg conversion and migration inhibition but also a proliferation of the CD25- subset. However, the PD-1$^{high}$CD25-Foxp3$^+$CD4 Tregs had minimal proliferation as assessed by Ki67 expression, suggesting Treg conversion and inhibition of Treg tumor egress rather than proliferation as the causes of accumulation in TILs. Whether the Treg conversion is a direct effect of anti-PD-1 on CD25$^+$Tregs or the PD-1$^{high}$CD25$^-$ Tregs are first targeted for migration inhibition which is then followed by conversion remains to be determined.

The preferential migration of TILs from primary tumors to dLNs via afferent lymphatic vessels was demonstrated by using photoconvertible Kaede transgenic mice, and the majority of these migrated T cells have an effector rather than regulatory phenotype[49], consistent with our observations here. We observed that CD80 blockade caused CD4 Teff accumulation in melanoma TILs and a concomitant decrease in dLNs, suggesting lymphatic migration was inhibited by blockade of CD80/PD-L1. CD80 blockade also promoted accumulation of T-bet$^+$ IFNγ-producing CD25$^+$CD4 Teffs, and reinvigorated exhausted CD8 in TILs. Reinvigorating the exhausted Teffs by blockade of the PD-1 pathway has proven efficacy in cancer therapy. In our study, PD-1 blockade not only reversed T cell exhaustion but also unleashed Teff immunity since IFNγ-producing PD-1$^{high}$ CD25$^-$ Tregs have less suppressive function[39,50]. Thus, the increased

frequencies of IFNγ-producing Tregs or CD4 Teffs, and granzyme B$^{high}$ CD8 Teffs in TILs by anti-PD-1 and anti-CD80 treatment may all have contributed to the melanoma regression, suggesting a potential combination therapy for melanoma with PD-1 and CD80 Abs. Anti-PD-1 combined with anti-cytotoxic T-lymphocyte-associated protein 4 (CTLA4) exhibits superior antitumor efficacy compared with single-agent therapy[51], suggesting that CTLA4 contributes to mechanisms of effector T cell exhaustion or anergy. Anti-CD80 (1G10) also blocks CD80 binding to CTLA4[30], which is constitutively expressed on Tregs and reportedly depletes CD80/CD86 from APCs via trans-endocytosis[36,52]. Whether CTLA4 also depletes Teff CD80, and whether Teff CD80 and Treg CTLA4 regulate Teff reinvigoration or migration, and hence tumor regression, need further investigation. Of note, the increased IFNγ in TILs could also promote PD-L1 expression and the apoptosis of LEC, which might affect Treg or Teff tumor egress. Further investigation will be required to assess the regulation of Tregs and Teffs in TILs by PD-L1-blockade. However, interpretation of results may be difficult since PD-L1 is widely expressed by these T cell subsets, by other lymphoid cells, and by non-hematopoietic endothelial cells. Overall, our study provides previously undescribed functions for PD-1 or CD80-driven PD-L1 signaling in endothelial cells for Treg or Teff migration and function. This information may improve the efficacy and strategies for therapies for autoimmune diseases and cancer.

## Methods

**Mice**. C57BL/6 J (CD45.2 and CD45.1) (female, 7–10 weeks old) were purchased from The Jackson Laboratory (Bar Harbor, ME). C57BL/6.Foxp3GFP mice were kindly provided by Dr. A. Rudensky (Memorial Sloan Kettering Cancer Center)[53]. PD-1$^{-/-}$ and PD-L1$^{-/-}$ mice have been described[2]. All animal care and experiments were carried out using protocols approved and overseen by the University of Maryland IACUC in compliance with state and federal guidelines.

**Antibodies and reagents**. Functional grade purified antibodies against mouse PD-1 (Rmp1-14), mouse PD-L1 (10 F.9G2), mouse CD80 (1G10) were purchased from BioXCell; F(ab')2 and Fab Abs were prepared with Pierce$^{TM}$ F(ab')2 Micro preparation kit (#44688) and Pierce$^{TM}$ Fab Micro preparation kit (#44685), respectively, the fragmented Abs were further purified and desalted with AminoLink Immobilization Kit (#43426) and polyacrylamide desalting columns (#43426). All kits were from Thermo Fisher Scientific, Inc. (Waltham, MA,). GoInVivo purified anti-human PD-1 (EH12.2H7), human PD-L1 (29E.2A3), and human PD-L1 (MIH3) were purchased from Biolegend. Recombinant mouse PD-1 Ig, PD-L1 Ig, CD80 Ig, CTLA4 Ig, human IgG1 or PE anti-human IgG1, and mouse IFNγ and TNFα were purchased from R&D Systems (Minneapolis, MN). All antibodies for flow cytometric analysis were purchased from Biolegend (San Diego, CA): Alexa Fluor® 647 anti-human PD-1 (EH12.2H7); APC anti-mouse PD-L1 (10 F.9G2); Brilliant Violet 421™ anti-human PD-L1 (29E.2A3); FITC anti-mouse CD80 (16-10A1); PE anti-human CD80 (2D10); APC anti-mouse VCAM-1 (429); Brilliant Violet 421™ or PerCP/Cyanine5.5 anti-mouse CD4 (RM4-5); APC anti-mouse CD25 (PC61); PE anti-mouse Foxp3 (MF-14); APC anti-mouse IFNγ (XMG1.2); FITC anti-mouse T-bet (4B10); APC anti-mouse Ki-67 (16A8); PerCP/Cyanine5.5 anti-mouse CD8 (53-6.7); APC anti-human/mouse Granzyme B (QA16A02); APC anti-mouse TIM-3 (B8.2C12); PE anti-mouse LAG-3 (C9B7W); except for PE anti-mouse PD-1 (J43, eBioscience) and PE anti-mouse Lyve-1 (ALY7, eBioscience). Antibodies for Immunohistochemistry: Purified Armenian Hamster anti-mouse PD-1 (J43) or CD80 (16-10A1) and rat anti-mouse PD-L1 (10F9G2), mouse VE-cadherin (11D4.1), or mouse VCAM-1 (429) were purchased from BD Biosciences

(Franklin Lakes, NJ); ZO-1 mouse mAb (ZO-1-1A12) and Foxp3 rat mAb (FJK-16s) from Thermo Fisher Scientific; eFluor 450 anti-mouse Lyve-1 (ALY7, eBioscience); anti-CD4 rabbit mAb (ab183685, Abcam, Cambridge, MA); Pacific Blue™ anti-mouse CD25 (PC61, Biolegend). NF-κB p65 (C22B4) rabbit mAb (#4764), F-actin-binding Alexa Fluor® 555 Phalloidin, and the antibodies against phosph-ERK1/2 (Thr202/Tyr 204), total ERK1/2, phospho-p65 (Ser536), phospho-Akt (Thr308), and GAPDH were obtained from Cell Signaling (San Diego, CA), NFκB-IκBα inhibitor BAY11-7082 and ERK inhibitor U0126 were purchased from Sigma–Aldrich (St. Louis, MO); PI3K p110α inhibitor PIK2 from Echelon Biosciences, Inc. (Salt Lake City, UT).

**Primary LECs and tumor cells**. Primary dermal LECs of C57BL/6 mouse (C57-6064L) or human (H-6064L) were from Cell Biologics, Inc. (Chicago, IL), and were cultured according to the manufacturer's instructions in manufacturer-provided mouse endothelial cell medium supplemented with 5% FBS, 2 mM L-glutamine, 100 IU /mL penicillin, vascular endothelial growth factor, endothelial cell growth supplement, heparin, epidermal growth factor, and hydrocortisone. Primary skin LECs were freshly isolated from the ears of wild-type C57BL/6 mice as previously described[25]. Briefly, ears were digested in 4 mg/ml collagenase D (Roche, Indianapolis, IN) at 37 °C for 1 h, washed, resuspended in mouse endothelial cell medium (Cell Biologics, Inc) and plated in six-well tissue culture plates overnight. The adherent cells were harvested for flow cytometry analysis. B16F10-Fluc/eGFP was purchased from Imanis Life Sciences (Rochester, MN), and maintained in DMEM supplemented with 10% FBS, 1× Penicillin/Streptomycin, 0.8 mg/mL G418, and 1 μg/mL puromycin.

**T cell subsets, B cells, and bone-marrow-derived dendritic cells**. CD4, CD8, and B cells from mouse LNs and spleens were isolated using CD4, CD8, and CD19 negative selection kits (Stemcell Technologies, Cambridge, MA), and were cultured as previously described[25]. Briefly CD4+CD25−Foxp3GFP− naïve T cells with >98% purity were sorted using a FACS Aria II (BD Biosciences, San Jose, CA). The sorted Foxp3GFP- naïve CD4 T cells or isolated CD8 T cells were then cultured for 3 days at 37 °C in 5% CO2, with IL-2 (20 ng/ ml, eBioscience), plate-bound anti-CD3ε mAb (1 μg/ mL, clone 145-2C11, eBioscience), and anti-CD28 mAb (1 μg/mL, clone 37.52, eBioscience) for activated Foxp3GFP-CD4 or CD8 T cells; and with recombinant human TGFβ1 (10 ng/ mL, eBioscience) and anti-mouse IL-4 (10 ng/mL, clone 11B11, eBioscience) for iTregs. Foxp3GFP−CD25+CD4+ Teffs, Foxp3GFP+CD25+CD44low CD4+ tTregs, or Foxp3GFP+CD25+CD44highCD4+ iTregs were FACS-sorted and cultured in RPMI 1640 supplemented with 10% FBS, 1 mM sodium pyruvate, 2 mM L-glutamine, 100 IU/mL penicillin, 100 μg /mL streptomycin, non-essential amino acids and 0.02 mM 2-ME (Sigma–Aldrich). The effector functions of sorted Foxp3GFP−CD25+CD4+Teffs were tested by phenotypic and functional analysis (Supplementary Fig. 1). Isolated B cells were activated with 10 μg/mL anti-IgM (Thermo Fisher) and 20 ng/mL anti-mouse IL-4 (eBioscience) for 3 days. Immature BMDCs were generated as described[54]. Briefly, bone marrow cells of wild-type mice were treated with 10 ng/ml GM-CSF (R&D Systems) for 10 days in petri dishes, and the loosely attached cells were collected. CD11c+ DC were purified by CD11c positive selection kit (Stemcell Technologies). Immature BMDCs were treated with 200 ng/mL LPS for 48 h to become mature BMDCs.

**CRISPR/Cas9 knockdown of PD-L1 in LECs**. The PD-L1 CRISPR guide RNA1 sequence: gtatggcagcaacgtcacga (disrupting exon1) was cloned into pLentiCRISPRv2 vector (a gift from Feng Zhang, Addgene plasmid # 52961) at the BsmBI site following the protocol described[55]. Lentivirus was produced by co-transfecting HEK293T cells with the packaging plasmids psPAX2 and pMD2.G (gifts from Didier Trono, Addgene plasmid # 12260 and 12259), and the transfer plasmid LentiCRISPRv2-gRNA PD-L1, using Lipofectamine 2000 (Thermo Fisher Scientific). The media was changed to antibiotic-free complete DMEM with 10% FBS after 16 h. The lentivirus supernatant was collected 24, 48, and 72 h after transfection and filtered through 0.45 μm PES syringe filter (SARSTEDT, Newton, NC). LECs were transduced with this lentivirus for 3 days, followed by a selection medium containing 2 μg/ml puromycin (Sigma–Aldrich) for 3 days. Surviving cells were expanded and FACS-sorted for Lyve1+PD-L1− LECs.

**Human T cell purification and culture**. Naïve human Tregs (CD4+CD25highCD127−CD45RA+) and naïve CD4 T cells (CD4+CD25-CD127+CD45RA+) were sorted via FACSAria from peripheral blood mononuclear cells. The sorted cells were incubated with an irradiated K562 cell line engineered to express CD86 and the high-affinity Fc Receptor (CD64) (KT86/64) as previously described[56]. Briefly, cells were cultured in XVivo-15 (BioWhittaker, Walkersville, MD) media containing 10% human AB serum (Valley Biomedical, Winchester, VA), Pen/Strep (Invitrogen, Carlsbad, CA), N-acetyl cysteine (USP), and recombinant IL-2 (300 IU/mL; Chiron, Emeryville, CA). After 14 days, cells were frozen. When needed, frozen naïve Tregs or CD4 T cells were thawed and restimulated with anti-CD3/CD28 mAb-Dynabeads (Life Technologies, Carlsbad, CA) at 1:3 (cell to bead) plus recombinant IL-2 (300 U/ml) for 10 days before assay.

**Cell viability and apoptosis assays**. For viability, LECs were treated as indicated for 72 h, washed, and incubated for 3 h with 0.5 mg/mL MTT (3-(4, 5-Dimethyl-thiazol-2-yl)-2, 5 diphenyl tetrazolium bromide) (Sigma–Aldrich). Fifty microliters of DMSO were added to cells before reading OD at 550 nm and 690 nm. For apoptosis, LECs were treated as indicated for 36 h, washed, and stained with PE Annexin V Apoptosis Detection kit (BD Biosciences) following the instructions.

**Flow cytometry**. Cells were incubated with antibodies for flow cytometry for 30 min at 4 °C, washed with PBS, and fixed with 4% paraformaldehyde, and run on an LSR Fortessa flow cytometer (BD Biosciences). For intracellular staining, cells were permeabilized with BD perm/fix buffer prior to incubation with antibodies. Results were analyzed with FlowJo 10.7.1 (Treestar).

**Immunoblotting**. Cells were lysed in buffer containing 20 mM Hepes (pH 7.4), 150 mM NaCl, 10 mM NaF, 2 mM Na3VO4, 1 mM EDTA, 1 mM EGTA, 0.5% Triton X-100, 0.1 mM DTT, 1 mM PMSF, and protease inhibitor cocktail (Roche). Protein in the cell extract was quantified using a protein quantification kit (Bio-Rad, Philadelphia, PA) and 10 μg total protein was run on Novex™ WedgeWell™ 4–20% Tris-Glycine Mini Gels (Invitrogen) and transferred to an Immobilon-P membrane (Bio-Rad). Membranes were probed with indicated antibodies. Relative band intensities of the blots were measured with ImageJ and normalized to GAPDH.

**Protein binding assay and co-immunoprecipitation**. Plated LECs were incubated with CD80 Ig, PD-1 Ig, or CTLA4 Ig (all fused to C-terminal human IgG1) or control human IgG1 for 1 hour at 37 °C. After washing, the LECs were either stained with mouse anti-human IgG1 (clone HP6069, Thermo Fisher) and rat anti-mouse PD-L1 (10F9G2) for immunohistochemistry, or stained with PE anti-human IgG1 for flow cytometry analysis of bound Igs. Alternatively, after washing, the LECs collected and lysed in the protein lysis buffer (see Immunoblotting method). The PD-L1-CD80 Ig immune complex was immunoprecipitated by sequential incubations and washes at 4 °C with 1 μg/mL anti-PD-L1 Ab (10F9G2) overnight and 30 μL of 50% protein A/G agarose slurry (Thermo Fischer) for 4 h, and then immunoblotted with mouse anti-human IgG1 (Clone HP6069, Thermo Fisher). The same whole-cell lysates were also immune blotted with anti-PD-L1. For Teff-PD-L1 Ig binding assay, $3 \times 10^5$ Teffs were incubated in a 96-well plate coated with 1 μg/mL PD-L1 Ig or human IgG1 for 3 h. MTT (0.5 mg/mL) was added 2 h before the plate reading.

**Transendothelial migration in vitro**. Transmigration across endothelial cells was described previously[27,57]. In brief, the inverted 5 μm pore size transwell insert (24-well, Corning International) was coated with 0.2% (w/v) gelatin (Bio-Rad) before loading with $1 \times 10^5$ LECs. Before migration, $2 \times 10^5$ migrating cells in 100 μL were loaded into the upper chamber of the transwell plate while the lower chamber contained 50 ng/mL mCCL19 (R&D systems), 100 ng/mL hCCL19 (R&D systems), 500 ng/mL mouse CXCL12 (R&D systems), 500 ng/mL mouse CCL21 (R&D systems), or 200 nM S1P (Sigma–Aldrich). All cells or reagents were prepared in IMDM containing transferrin and 0.5% (w/v) fatty acid-free BSA (Gemini, West Sacramento, CA). T cells that migrated to the lower chamber after 3 h at 37 °C were counted.

**Time-lapse microscopy**. Cfse-labeled, FACS-sorted WT, PD-1−/−, PD-L1−/− iTreg cells ($5 \times 10^4$ cells per transwell) migrating across endothelial monolayers to CCL19 (50 ng/ mL) were visualized by EVOS FL Auto Cell Imaging System (Thermo Fisher Scientific) with a ×20 objective. One image was captured every 5 min for 3 h. Cell tracks were analyzed with Volocity version 6.3 software (Perkin Elmer).

**Immunohistochemistry**. Cell monolayers or tissues were fixed for 20 min at 4 °C with 4% (w/v) paraformaldehyde (Affymetrix, Santa Clara, CA), then permeabilized with PBS 0.2% (v/v) Triton X-100 (Sigma–Aldrich), and treated with 4% donkey serum for 30 min then incubated with primary antibodies for overnight at 4 °C. The bound antibodies were detected with Alexa Fluor 448, 647 (Cy5), or 546 (Cy3)-conjugated secondary antibodies (Jackson ImmunoResearch, West Grove, PA) for 1 h at 4 °C. The mounted slides were visualized by fluorescent microscopy (Zeiss LSM 510 Meta and LSM5 Duo). The mean fluorescence intensity (MFI) of images and the T cell PD-1 or CD80 colocalization with PD-L1 on LEC were analyzed with Volocity version 6.3 software. Quantification of the junctional VE-cadherin in ×60 magnified images of LVs of whole mounted LVs or adherent LECs was performed with ImageJ. Length of zipper junctions and button junctions were measured. The percentage of zipper junction was calculated as: length of zipper junction × 100 / (length of zipper junction + length of button junction)[31].

**Footpad migration assay**. Mice were anaesthetized, and $1 \times 10^6$ CFSE-labeled Tregs or non-Tregs were injected intradermally into the footpads in 20 μL PBS as we previously described. Draining popliteal LN were collected 12 h post-injection and processed for flow cytometry.

**Tumor treatments.** C57/BL6J (CD45.2) mice were subcutaneously injected with $1 \times 10^6$ B16F10-fluc/eGFP tumor cells. Seven and Ten days after tumor inoculation, tumor-bearing mice were injected intraperitoneally with anti-PD-1 (Rmp1-14), anti-CD80 (1G10), or rat IgG2a (2A3). Two days after the last antibody treatment, 5 mice from each group were euthanized, and the tumors, dLNs, and non-dLNs were harvested and analyzed by flow cytometry and immunohistochemistry. At 8 days after B16F10 tumor inoculation, tumor-bearing mice had an intratumoral injection of $0.5 \times 10^6$ FACS-sorted iTregs cultured from naïve CD4 T cells of Foxp3GFP C57BL/6 (CD45.1) mice. These iTregs were pretreated with 2 µg/mL anti-PD-1 (Rmp-14) or the rat IgG (2A3). Sixteen hours after Treg transfer, tumors and dLNs (4 mice/group) were analyzed by flow cytometry. Tumor volume in the parallel groups of mice (7–8 mice in each group) was monitored daily with an analysis threshold for survival set at 0.25 cm³.

**Statistical analysis.** Numerical data are presented as mean ± SEM. Asterisks mark data statistically different from the controls, with *p*-values noted in the figure legends. A *p*-value of <0.05 was considered significant for one-way ANOVA and unpaired, two-tailed *t*-tests using Prizm 8 software. The number of replicates is noted in the figure legends.

**Reporting summary.** Further information on experimental design is available in the Nature Research Reporting Summary linked to this paper.

## Data availability

The authors declare that the data supporting the findings of this study are available within the paper and its supplementary information files. Source data are provided with this paper.

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

## Acknowledgements

This work was supported by NIH grants R37AI062765, R01 AI114496, and P01 AI153003 to J.S.B.; RO1 HL155114 and R37AI34495 to B.R.B.; R01 HL11879 and P01 CA 065493 to B.R.B. and K.L.H. We thank Dr. Xiaoxuan Fan, Bryan Han, and Karen Underwood from UMB-Flow Core Facility for excellent help for flow cell sorting.

## Author contributions

W.P. and J.S.B. designed the research. W.P., L.L., Y.Z., V.S., J.I., R.L., E.S., and C.P. performed the experiments. Y.Z., K.L.H., and B.R.B. supplied human T cells. I.T.L. and L.V.R. supplied PD-1 and PD-L1 knockout mice. Y.L. helped with tumor inoculations. W.P. and J.S.B. analyzed the results and wrote the manuscript.

## Competing interests

The authors declare no competing interests.
