## [Peer Review File · Nature Communications]

PD-L1 signaling selectively regulates T cell lymphatic transendothelial migrationREVIEWER COMMENTS

Reviewer #1 (Remarks to the Author):

In this paper, the authors focused on the role of PD-L1 on lymphatic endothelial cells (LECs) in transendothelial migration (TEM). They found a preferentially dominant contribution of PD-1 signaling for iTreg and CD80 signaling for Teff to TEM mediated by PD-L1-expressing LECs using by transendothelial migration assay in vitro and labelled cell-transferred foodpad migration assay and B16F10 tumor inoculation assay in vivo. They also found the differential signaling of ERK, NF- κ B, and Akt between PD-1/PD-L1 and CD8/PD-L1 pathways. Most experiments have been well-designed and performed in suitable ways. There are several concerns in the present form of this manuscript.

Major concerns

1. They presented the expression levels of PD-1 and PD-L1 (MFI) in the same panel (Fig.1 a,b,c left panels). This is the results from the use of different mAbs. They cannot directly compare the MFI levels. They should present the differences of MFI between PD-1 and PD-L1 expression in each cell subset and should present as delta (Δ)MFI. In the text, they also compared MFI of PD-1 and PD-L1 and described that "PD-1 is higher than PD-L1" (p4-5). They need to restate the related all parts.
2. In the B16F10 tumor inoculation experiment, they inoculated tumor cells 7 days before, and treated with either anti-PD-1 or anti-CD80 mAb at day 0 and 3, and then analyzed tumors and draining LNs at day 7. This experimental design cannot directly investigate migration of iTreg and Teff to tumors and dLN. I suggest to perform cell transfer of CD25+Foxp3+CD4+ Treg cells into B16F10-bearing immunocompromised host mice to see direct migration and possible conversion to CD25+ Tregs by PD-1 blockade.

Minor concerns

1. They abbreviated "activated conventional CD4+ T cells" as "Teffs". I think that it is better to use activated Tcon (act Tcon) instead of Teffs, because they have not observed any effector function of activated CD4+ T cells. This naming confuses us.
2. In the first appearance of tumor (melanoma), they only mentioned "melanoma", but they need to state a actual cell line name (p12, line252).

Reviewer #2 (Remarks to the Author):

In the manuscript titled "PD-L1 signaling selectively regulates lymphatic transendothelial migration" the authors describe a mechanism by which iTregs and Teff bind to PD-L1 via PD-1 or CD80 to initiate a signaling cascade within the LECs that leads to down regulation of VE-Cadherin and upregulation of VCAM1 expression by LECs. In turn the LECs assist in transmigration based on the interaction between PD-1 on iTreg and PD-L1 on LEC or CD80 on Teff and PD-L1 on LEC. The authors suggest that blocking these interactions prevents tumor egress and promotes tumor regression. This is an interesting study that builds on recent findings about cis and trans interactions between PD-1, CD80 and PD-L1 as well as studies demonstrating PD-L1 expression by LECs. However, the authors fail to fully

appreciate the breadth of literature around these interactions, the unique features of LECs in PD-L1 expression, upregulation, reverse signaling, and the different types of cell-cell junctions found within the lymphatic vasculature. A much stronger manuscript would result from additional experiments to address the current literature and additional referencing as noted below.

Major Criticisms.

1. While the authors conclude that changes in VE-Cadherin expression may be the cause of differences in TEM they fail to fully demonstrate that this is the case. VE-Cadherin is a critical marker of button-like versus zipper-like junctions in LECs (references to include: PMID: 17846148, 32546411, 30093598) that lead to changes in vascular permeability. While expression of VE-Cadherin may regulate vascular permeability, in lymphatic endothelial cells it is the reorganization of the VE-Cadherin that is important for permeability. Some quantification of changes in VE-Cadherin localization and relationship to upregulation of VE-Cadherin should be demonstrated in order to conclude that changes in VE-Cadherin are responsible for the changes observed in migration.
2. The authors fail to address the relevance of PD-L1 expression by lymphatic endothelial cells and any published data regarding PD-L1 reverse signaling. As an example, lymphatic endothelial cells proliferate more in the absence of PD-L1 and undergo more apoptosis (PMID: 30045970) similar to cancer cells (PMID: 18223165) and is significantly upregulated in the presence of type 1 IFN, TNF, type 2 IFN, etc. Further, expression of PD-L1 by in vitro primary cells is significantly lower than in LN LECs and the expression of PD-L1 by tissue lymphatics in homeostasis is fairly low, however upregulation has been documented both during infection and in a tumor setting (PMID:30381467 among others). Some demonstration of how TEM is affected by PD-L1 upregulation in the presence of type 1 IFN, TNF or Type 2 IFN would significantly improve the interpretations within the manuscript. Additionally, evaluation of viability and proliferation of the PD-L1-/- LECs is required based on the known role for PD-L1 in conferring survival and proliferation differences.
3. There was significant oversight when addressing the trans binding of CD80 to PD-L1. While one manuscript was cited regarding the interaction of CD80 with PD-L1 occurring only in cis (Ref 34: Chaudhri A et al Cancer immunology Research 2018) the authors conclude that in this situation the proteins bind in trans. In another (not cited) recent manuscript it was validated that CD80 and PD-L1 only interact in cis using membrane adhesion assays, total internal reflection of fluorescence (TIRF) as well as forster resonance energy transfer (FRET) (PMID: 31757674). As the authors conclude that PD-L1 and CD80 interact in trans in the submitted manuscript they must include more robust imaging/staining to demonstrate that this is the case. As the staining in figure 5 is inconclusive at best their conclusions about Teff are significantly over-stated and unacceptable in this state. This reviewer would recommend either including additional binding evidence that in these cell types CD80 and PD-L1 interact in trans or focus only on the iTreg portion of the story as the Teff data is tenuous.

Minor Criticisms.

1. In figure 1c inclusion of levels of PD-1, PD-L1 and CD80 from LECs in vivo using immunofluorescence or flow cytometry from cells isolated from tissue or lymph node in comparison to primary cells purchased from cell biologics would help to demonstrate the actual amount of PD-L1 on LECs and determine if the purchased LECs actually have "high levels" of PD-L1. Expression of PD-L1 could also be compared to another cell type not expressing PD-L1.
2. In figure 2 some quantification of numbers of cells migrated should be displayed rather

than percent of cells.

3. In figure 2 they demonstrate that PD-L1 Ig induces iTreg migration. CD80 Ig should also be included in this figure.

4. In figure 2 western blot loading controls used for normalization (GAPDH and ERK) are either overexposed or uneven. Please provide images with lower exposure.

5. The quality of the immunofluorescence staining in figure 3 e, g (and others) is poor and larger, higher resolution images (in focus) with zoomed in insets are required to be able to visualize staining.

6. Also, for the immunofluorescence in Figure 3 the antibody manufacturer shows ZO-1 staining to be at Cell-Cell junctions. It is difficult to determine if this is an issue of image quality, as mentioned above, or if the ZO-1 in 3e and g isn't actually staining cell-cell junctions.

7. In figure 3 the antibody labels are inaccurate with the text and need to be changed. Is 2F9G2 actually 10F.9G2? Is MIH3 actually EH12.H7?

8. In figure 4 some discussion and quantification of button versus zipper junctions should be included (see main point 1).

9. It is unclear why CFSE labeled cells in figure 5 are not dapi positive.

10. In figure 5, if CD80/PD-L1 co-localization in trans is going to be included, FRET or similar methodologies should be used to quantify and demonstrate this finding. Additional data showing colocalization with Pdl1^{-/-} LECs, antibody treated (PD-L1, PD-1 or CD80) and recombinant proteins should be included.

11. Figure 6, much of the previous data demonstrates what happens with anti-PD-L1. Why was anti-PD-L1 not included here as well?

12. Figure 7, "PD-1 blockade prevents afferent lymphatic migration of CD25-Foxp3+ CD4 Tregs from TILs to the dLNs" is the title, but no demonstration of lymphatic migration or visualization of the lymphatic vasculature is shown. Thus, without directly showing fewer cells in the lymphatic vasculature over time the title needs to be re-worded or the data needs to be added.

13. Figure 7a, again images are poor and we need a zoomed in example. Additionally, quantification appears to be MFI of each stain, but perhaps number of cells would be more representative. Further, some quantification by flow cytometry of number and MFI could be demonstrated to strengthen the data in A.

14. As CTLA4 expressed by Tregs can also bind to CD80 (PMID: 31757674) some discussion of CTLA4 is required.

15. There are also reports demonstrating anti-PD-L1 may be agonistic. Please address whether anti-PD-L1 could be agonistic in this situation and if this could affect the TEM.

REVIEWER COMMENTS

Reviewer #1 (Remarks to the Author):

In this paper, the authors focused on the role of PD-L1 on lymphatic endothelial cells (LECs) in transendothelial migration (TEM). They found a preferentially dominant contribution of PD-1 signaling for iTreg and CD80 signaling for Teff to TEM mediated by PD-L1-expressing LECs using by transendothelial migration assay in vitro and labelled cell-transferred foodpad migration assay and B16F10 tumor inoculation assay in vivo. They also found the differential signaling of ERK, NF-kB, and Akt between PD-1/PD-L1 and CD8/PD-L1 pathways. Most experiments have been well-designed and performed in suitable ways. There are several concerns in the present form of this manuscript.

Major concerns

1. They presented the expression levels of PD-1 and PD-L1 (MFI) in the same panel (Fig.1 a,b,c left panels). This is the results from the use of different mAbs. They cannot directly compare the MFI levels. They should present the differences of MFI between PD-1 and PD-L1 expression in each cell subset and should present as delta (Δ) MFI. In the text, they also compared MFI of PD-1 and PD-L1 and described that “PD-1 is higher than PD-L1” (p4-5). They need to restate the related all parts.

The MFI levels of PD-1 and PD-L1 expression are now separated for each cell subset and presented as delta (Δ) MFI (Fig. 1a, b, left panels). The remark that “PD-1 is higher than PD-L1” and related part are now deleted in line 88-89.

2. In the B16F10 tumor inoculation experiment, they inoculated tumor cells 7 days before, and treated with either anti-PD-1 or anti-CD80 mAb at day 0 and 3, and then analyzed tumors and draining LNs at day 7. This experimental design cannot directly investigate migration of iTreg and Teff to tumors and dLN. I suggest to perform cell transfer of CD25⁺Foxp3⁺CD4⁺ Treg cells into B16F10-bearing immunocompromised host mice to see direct migration and possible conversion to CD25⁺ Tregs by PD-1 blockade.

We agree with the suggestion and performed intratumoral cell transfer of CD25⁺Foxp3GFP⁺CD4⁺ iTregs into B16F10-bearing hosts to directly assess migration and Treg conversion. iTregs were isolated and cultured from Foxp3GFP C57BL/6 (CD45.1) mice, and then treated with anti-PD-1 Ab (Rmp1-14) or the isotype rat IgG2a (2A3). These cells were then injected into tumors of C57BL/6 (CD45.2) mice 8 days after B16F10 inoculation. 16 hours later the tumors and draining LNs were analyzed. PD-1 blockade of iTregs reduced their migration to the dLNs and increased their accumulation in the tumor.

In addition, the majority of the retained Tregs were CD25⁻ and IFN γ ^{high}, suggesting parallel events of impaired Treg migration and increased Treg conversion by PD-1 blockade. These results are now included in lines 326- 336, Fig. 7e-g, and discussed in lines 411-413.

Minor concerns

1. They abbreviated “activated conventional CD4⁺ T cells” as “Teffs”. I think that it is better to use activated Tcon (act Tcon) instead of Teffs, because they have not observed any effector function of activated CD4⁺ T cells. This naming confuses us.

We tested the effector functions of the activated CD4 T cells cultured from the FACS-sorted Foxp3GFP⁺CD4 T cells. These cells are IFN γ -producing CD44^{high}CD45RB⁺ CD62L⁻

T-bet⁺, and are cytotoxic to B16F10 melanoma cells. We now include these results in Supplementary Fig. 1, and therefore have kept the “Teff” abbreviation.

2. In the first appearance of tumor (melanoma), they only mentioned “melanoma”, but they need to state a actual cell line name (p12, line252).

The error is now corrected as “B16F10 melanoma” in line 285 (p13)

Reviewer #2 (Remarks to the Author):

In the manuscript titled “PD-L1 signaling selectively regulates lymphatic transendothelial migration” the authors describe a mechanism by which iTregs and Teff bind to PD-L1 via PD-1 or CD80 to initiate a signaling cascade within the LECs that leads to down regulation of VE-Cadherin and upregulation of VCAM1 expression by LECs. In turn the LECs assist in transmigration based on the interaction between PD-1 on iTreg and PD-L1 on LEC or CD80 on Teff and PD-L1 on LEC. The authors suggest that blocking these interactions prevents tumor egress and promotes tumor regression. This is an interesting study that builds on recent findings about cis and trans interactions between PD-1, CD80 and PD-L1 as well as studies demonstrating PD-L1 expression by LECs. However, the authors fail to fully appreciate the breadth of literature around these interactions, the unique features of LECs in PD-L1 expression, upregulation, reverse signaling, and the different types of cell-cell junctions found within the lymphatic vasculature. A much stronger manuscript would result from additional experiments to address the current literature and additional referencing as noted below.

Major Criticisms.

1. While the authors conclude that changes in VE-Cadherin expression may be the cause of differences in TEM they fail to fully demonstrate that this is the case. VE-Cadherin is a critical marker of button-like versus zipper-like junctions in LECs (references to include: **PMID: 17846148, 32546411, 30093598**) that lead to changes in vascular permeability. While expression of VE-Cadherin may regulate vascular permeability, in lymphatic endothelial cells it is the reorganization of the VE-Cadherin that is important for permeability. Some quantification of changes in VE-Cadherin localization and relationship to upregulation of VE-Cadherin should be demonstrated in order to conclude that changes in VE-Cadherin are responsible for the changes observed in migration.

We agree that VE-Cadherin is a critical marker of button and zipper-like junctions in LECs. We now acknowledge that reorganization of VE-cadherin around LEC is important for permeability. We have now quantified the zipper junctions which are upregulated in PD-L1 KO LVs (Fig. 3g) and LECs (Fig. 3h, Fig. 4b, c, Supplementary Fig. 4a). We now discuss this regulation in the Results (lines 182-184, 189-191, 219-225) and Discussion (lines 377; 392-395) with the supporting references.

2. The authors fail to address the relevance of PD-L1 expression by lymphatic endothelial cells and any published data regarding PD-L1 reverse signaling. As an example, lymphatic endothelial cells proliferate more in the absence of PD-L1 and undergo more apoptosis (**PMID: 30045970**) similar to cancer cells (**PMID: 18223165**) and is significantly upregulated in the presence of type 1 IFN, TNF, type 2 IFN, etc. Further, expression of PD-L1 by in vitro primary cells is significantly lower than in LN LECs and the expression of PD-L1 by tissue lymphatics in homeostasis is fairly low, however upregulation has been documented both during infection and in a tumor setting (**PMID:30381467** among others). Some demonstration of how TEM is affected by PD-L1

upregulation in the presence of type 1 IFN, TNF or Type 2 IFN would significantly improve the interpretations within the manuscript. Additionally, evaluation of viability and proliferation of the PD-L1^{-/-} LECs is required based on the known role for PD-L1 in conferring survival and proliferation differences.

Regarding PD-L1 expression in primary LECs, we isolated fresh dermal LECs and detected levels of PD-L1 expression comparable to the commercially purchased dermal LECs (included in Fig 1c; lines 101-105). We also analyzed WT and PD-L1 KO (CRISPR/Cas9) LEC apoptosis and viability with or without the influence of IFN γ or TNF α . IFN γ induced WT LEC apoptosis after 36 hours incubation, while it failed to induce apoptosis in PD-L1^{-/-}LECs. 72 hours cultured PD-L1^{-/-} LECs had slightly better survival than wild type LECs. These results are included in Supplementary Fig. 4e, f; lines 193-196 with the noted references. We observed that IFN γ enhanced PD-L1 expression on LECs, and the increased PD-L1 expression promoted Treg TEM (included in Fig. 3i; lines 196-197). We also discussed the data in lines 455-458.

3. There was significant oversight when addressing the trans binding of CD80 to PD-L1. While one manuscript was cited regarding the interaction of CD80 with PD-L1 occurring only in cis (**Ref 34: Chaudhri A et al Cancer immunology Research 2018; used 300.19 B cell lymphoma, EL-4 T cell lymphoma, and COS cells along with recombinant proteins.**) the authors conclude that in this situation the proteins bind in trans. In another (not cited) recent manuscript it was validated that CD80 and PD-L1 only interact in cis using membrane adhesion assays, total internal reflection of fluorescence (TIRF) as well as forster resonance energy transfer (FRET) (**PMID: 31757674; used HEK, Jurkat, Raji, OT-1, and TIL DM and M Φ along with recombinant proteins**). As the authors conclude that PD-L1 and CD80 interact in trans in the submitted manuscript they must include more robust imaging/staining to demonstrate that this is the case. As the staining in figure 5 is inconclusive at best their conclusions about Teff are significantly over-stated and unacceptable in this state. This reviewer would recommend either including additional binding evidence that in these cell types CD80 and PD-L1 interact in trans or focus only on the iTreg portion of the story as the Teff data is tenuous.

Regarding the interaction of CD80 with PD-L1 occurring in cis, we agree with this possibility in T cells and APCs which express both CD80 and PD-L1. We propose the hypothesis in Supplementary Fig. 6 where we consider the co-existence of both cis and trans binding of CD80 to PD-L1. In our models, LECs only express PD-L1 with no CD80 or PD-1 surface expression. Our evidence for CD80 binding in trans to PD-L1 on LEC includes the following: (i) CD80 Ig induced strong ERK activation in wild type but not PD-L1 KO LECs (Supplementary Fig. 4e) which was non-overlapping with PD-1 Ig induced signaling (predominantly AKT activation) (Fig. 3i). (ii) Masking CD80 on Teffs or blocking PD-L1 on LEC with specific PD-L1 antibody (10F.2H11), which solely blocks the PD-L1 and CD80 interaction, inhibited only Teff TEM (Fig. 3a). (iii) To prove trans binding of CD80/PD-L1, we performed binding assays using CD80 Ig to treat LEC, knowing that LECs have no surface CD80 or PD-1 expression. In the first set of experiments, LECs were incubated with CD80 Ig, PD-1 Ig, CTLA4 Ig, or control human IgG1; and we detected the PD-L1 bound fusion proteins with anti-human IgG1 by immunohistochemistry and flow cytometry. CD80 Ig and PD-1 Ig but not CTLA4 Ig bound to LEC PD-L1. No immunoglobulin fusion protein binding was observed on PD-L1^{-/-} LECs. The data are added as Fig. 5f-h. In the second set of experiments, after CD80 Ig or human IgG1 treatment, the LECs were collected and lysed after the unbound fusion proteins were removed. The PD-L1-CD80 complex was pulled down by anti-PDL1 mAb and Protein A/G agarose beads by immunoprecipitation and blotted. There was clear pull down of bound CD80 Ig but not human IgG1. The data are included as Fig. 5i. (iv) Our assays exclusively used primary LECs and T cells, and thus differed from the reports in the literature which relied on engineered tumor cell lines with

molecule over expression and focused on other cell types (PMID: 31757674; Ref. 34). The cis or trans interaction likely depends on the spatio-temporal dynamics of surface expression and cell types with unique functions. (v) Teffs bound to immobilized PD-L1 Ig, which was blocked by masking PD-L1 on PD-L1 Ig or CD80 on Teffs (Fig. 5ji). Further, PD-L1 Ig coated on the Boyden chamber increased Teff TEM, which was blocked by anti-CD80 (1G10). Masking PD-L1 Ig with anti-PD-L1 (10F.9G2) also blocked Teff TEM (Fig. 5jii). (vi) Incubating Teffs with increasing doses of CD80 Ig, did not alter PD-L1 expression (Supplementary Fig. 4h). (vii) Anti-PD-L1 treatment of Teffs did not affect TEM (Fig. 2a, b). Taken together, these data argue against an exclusive cis interaction of CD80/PD-L1 on Teffs during the interaction with LEC for TEM. We included the new results in lines 250-263, discussed them in lines 350-368, and cited the noted references.

Minor Criticisms.

1. In figure 1c inclusion of levels of PD-1, PD-L1 and CD80 from LECs in vivo using immunofluorescence or flow cytometry from cells isolated from tissue or lymph node in comparison to primary cells purchased from cell biologics would help to demonstrate the actual amount of PD-L1 on LECs and determine if the purchased LECs actually have "high levels" of PD-L1. Expression of PD-L1 could also be compared to another cell type not expressing PD-L1.
We performed flow cytometry analysis of PD-L1 expression on freshly isolated Lyve-1⁺CD31⁺ primary LECs or Lyve-1⁻CD31⁻ primary non-LECs vs purchased and cultured LECs. There is comparable expression of PD-L1 between fresh dermal LECs and cultured dermal LECs. These data are now included in lines 101-105 and Fig. 1c.
2. In figure 2 some quantification of numbers of cells migrated should be displayed rather than percent of cells.
Per Reviewer#2's suggestion, the percent of cells migrated has been replaced with the numbers of cells migrated throughout the manuscript.
3. In figure 2 they demonstrate that PD-L1 Ig induces iTreg migration. CD80 Ig should also be included in this figure.
We added the CD80 Ig experiment to Fig. 2i. There is no effect of CD80 Ig on iTreg transmigration (line 148).
4. In figure 2 western blot loading controls used for normalization (GAPDH and ERK) are either overexposed or uneven. Please provide images with lower exposure.
We replaced the GAPDH and ERK with lower exposures in Fig. 2i and j.
5. The quality of the immunofluorescence staining in figure 3 e, g (and others) is poor and larger, higher resolution images (in focus) with zoomed in insets are required to be able to visualize staining.
We replaced these with better images, including higher magnification insets in Fig. 3g and h.
6. Also, for the immunofluorescence in Figure 3 the antibody manufacturer shows ZO-1 staining to be at Cell-Cell junctions. It is difficult to determine if this is an issue of image quality, as mentioned above, or if the ZO-1 in 3e and g isn't actually staining cell-cell junctions.

We performed the IHC with newly purchased ZO-1 antibody and replaced Fig. 3e and h with higher quality images.

7. In figure 3 the antibody labels are inaccurate with the text and need to be changed. Is 2F9G2 actually 10F.9G2? Is MIH3 actually EH12.H7?
These errors are now corrected.
8. In figure 4 some discussion and quantification of button versus zipper junctions should be included (see main point 1).
We very much appreciate reviewer #2's suggestions about button versus zipper junctions. We quantified zipper-like junctions in Fig. 4b and 4c, and included these in the Results (lines 219-225) and Discussion (lines 392-395).
9. It is unclear why CFSE labeled cells in figure 5 are not dapi positive.
We replaced this with new images of Tregs and Teffs with DAPI in Fig. 5e.
10. In figure 5, if CD80/PD-L1 co-localization in trans is going to be included, FRET or similar methodologies should be used to quantify and demonstrate this finding. Additional data showing colocalization with Pd11-/- LECs, antibody treated (PD-L1, PD-1 or CD80) and recombinant proteins should be included.
We used several approaches to demonstrate the trans binding of CD80 to PD-L1; please see responses in major point #3 above.
11. Figure 6, much of the previous data demonstrates what happens with anti-PD-L1. Why was anti-PD-L1 not included here as well?
We considered assessing the regulation of Tregs and Teffs in TILs by PD-L1-blockade. However, interpretation of the results is not straight forward since PD-L1 is widely expressed not only by these T cell subsets, but also by other lymphoid cells and by non-hematopoietic endothelial cells. This is now discussed in lines 455-458.
12. Figure 7, "PD-1 blockade prevents afferent lymphatic migration of CD25-Foxp3+ CD4 Tregs from TILs to the dLNs" is the title, but no demonstration of lymphatic migration or visualization of the lymphatic vasculature is shown. Thus, without directly showing fewer cells in the lymphatic vasculature over time the title needs to be re-worded or the data needs to be added.
We performed intratumoral cell transfer of CD25⁺Foxp3GFP⁺CD4⁺ iTreg cells into B16F10-bearing mice to directly assess migration and Treg conversion. The transferred iTregs were isolated and cultured from Foxp3GFP C57BL/6 (CD45.1) mice, and then treated with anti-PD-1 Ab (Rmp1-14) or the isotype rat IgG2a (2A3). The cells were then transferred to the tumors of B16F10-bearing C57BL/6 (CD45.2) mice 8 days after B16F10 inoculation. 16 hours later the tumors and draining LNs were analyzed. PD-1 blockade of iTregs reduced migration to the dLNs and increased their accumulation in the tumor. In addition, a majority of the retained Tregs were CD25⁻ and IFN γ ^{high}, suggesting parallel events of impaired Treg migration and increased Treg conversion by PD-1 blockade. These results are now included in lines 326-336 and Fig. 7e-g. Based on these results, we re-titled Fig. 7 as: "PD-1 blockade prevents afferent lymphatic migration of CD25⁺Foxp3⁺CD4 Tregs from TILs to the dLNs and increases Treg conversion to CD25⁻ IFN γ ^{high} Tregs".

13. Figure 7a, again images are poor and we need a zoomed in example. Additionally, quantification appears to be MFI of each stain, but perhaps number of cells would be more representative. Further, some quantification by flow cytometry of number and MFI could be demonstrated to strengthen the data in A.

We replaced this with better quality images along with zoomed images. We used sum of mean MFI to represent the infiltrated cell quantity, which is more accurate than manual cell counting. Cell counts of Foxp3+/CD25+/CD4+ cells in TILs and dLNs by FACS are presented in Fig. 6f and g.

14. As CTLA4 expressed by Tregs can also bind to CD80 (PMID: 31757674) some discussion of CTLA4 is required.

This is good point, and we added the following to the Discussion, lines 447-457: “Anti-CD80 (1G10) also blocks CD80 binding to CTLA4 (Ref. 30), which is constitutively expressed on Tregs, and reportedly depletes CD80/CD86 from APCs via trans-endocytosis (Ref. 36, 50). Whether CTLA4 also depletes Teff CD80, and whether the interaction of Teff CD80 and Treg CTLA4 regulate Teff reinvigoration or migration, and hence tumor regression, need further investigation”.

15. There are also reports demonstrating anti-PD-L1 may be agonistic. Please address whether anti-PD-L1 could be agonistic in this situation and if this could affect the TEM.

In our models, anti-PD-L1 is not agonistic: (i) On T cells, anti-PD-L1 (10F9G2) treatment of Tregs or Teffs had no effect on migration (line 115, Fig. 2a; line 116, Fig. 2b). (ii) Anti-PD-L1 (10F9G2) treatment of LEC blocks T cell migration (lines 166-177, Fig.3a-c). Thus, we tested the possible agonistic activity of anti-PD-L1 (10F9G2) in LEC by immunoblotting. No specific signaling was induced in LECs by 10F9G2 (lines 197-202, Supplementary Fig. 4g).

REVIEWER COMMENTS

Reviewer #1 (Remarks to the Author):

Now the revised manuscript has been adequately corrected and responded to the comments to reviews. I think that this manuscript is acceptable for Nature Communication.

Reviewer #2 (Remarks to the Author):

Dear authors, we appreciate the responses to our comments and only have a few minor revisions that remain.

1. Lines 195-196 the authors state : "Consistent with previous reports, loss of PD-L1 is LEC increased cell viability and prevented apoptosis induced by IFN γ but not TNF α ." Please reword this sentence as this is incorrect. The previous reports demonstrate that loss of PD-L1 promotes apoptosis and PD-L1 actually protects against apoptosis.

2. In figure 5 dotted lines outlining the border of the cells in D&E would be helpful.

3. In figure 5 D&E some measurement of co-localization should be included. % cells with CD80 at the interface between WT PD-L1 LEC and KO PD-L1 LEC so that we can get an idea of the frequency of this event. This will also increase the rigor of these studies.

REVIEWERS' COMMENTS

Reviewer #1 (Remarks to the Author):

Now the revised manuscript has been adequately corrected and responded to the comments to reviews. I think that this manuscript is acceptable for Nature Communication.

Thanks Reviewer #1.

Reviewer #2 (Remarks to the Author):

Dear authors, we appreciate the responses to our comments and only have a few minor revisions that remain.

1. Lines 195-196 the authors state: "Consistent with previous reports, loss of PD-L1 in LEC increased cell viability and prevented apoptosis induced by IFN γ but not TNF α ." Please re-word this sentence as this is incorrect. The previous reports demonstrate that loss of PD-L1 promotes apoptosis and PD-L1 actually protects against apoptosis.

Yes, we corrected the statement, and reworded the sentences as bellow (now in lines 193-196):

"Previous reports ^{22, 33, 34}demonstrated that PD-L1 protects LN LECs and tumor cells from apoptosis, while we observed that loss of PD-L1 in dermal LEC increased cell viability and prevented apoptosis induced by IFN γ but not TNF α ."

2. In figure 5 dotted lines outlining the border of the cells in D&E would be helpful.

We thank for the suggestion and outline the border of the interacting cells with dot line in figure 5d and 5e.

3. In figure 5 D&E some measurement of co-localization should be included. % cells with CD80 at the interface between WT PD-L1 LEC and KO PD-L1 LEC so that we can get an idea of the frequency of this event. This will also increase the rigor of these studies.

We agree with the suggestions. We analyzed T cell and LEC colocalization by calculating Pearson's coefficient and Colocalization coefficient, using Volocity software. We also measured the percentage of CD80 at the interface between Teff-WT LEC and Teff-PD-L1KO LECs. These data are added into the figure 5d and 5e.